# Ground Slow, Move Fast: A Dual-System Foundation Model for Generalizable Vision-and-Language Navigation

**Meng Wei**[1,2] **Chenyang Wan**[1,3] **Jiaqi Peng**[1,4] **Xiqian Yu**[1] **Yuqiang Yang**[1] **Delin Feng**[1]
**Wenzhe Cai**[1] **Chenming Zhu**[1,2] **Tai Wang**[1,†] **Jiangmiao Pang**[1,‡] **Xihui Liu**[2,‡]

[1]Shanghai AI Laboratory    [2]The University of Hong Kong
[3]Zhejiang University    [4]Tsinghua University
[†] Project Lead [‡] Corresponding authors

🐙 **Code:InternNav** 📦 **Model:InternVLA-N1** 🗄 **Data:InternData-N1** 🏠 **Homepage**

## Abstract

While recent large vision-language models (VLMs) have improved generalization in vision-language navigation (VLN), existing methods typically rely on end-to-end pipelines that map vision-language inputs directly to short-horizon discrete actions. Such designs often produce fragmented motions, incur high latency, and struggle with real-world challenges like dynamic obstacle avoidance. We propose DualVLN, the first dual-system VLN foundation model that synergistically integrates high-level reasoning with low-level action execution. System 2, a VLM-based global planner, "grounds slowly" by predicting mid-term waypoint goals via image-grounded reasoning. System 1, a lightweight, multi-modal conditioning Diffusion Transformer policy, "moves fast" by leveraging both explicit pixel goals and latent features from System 2 to generate smooth and accurate trajectories. The dual-system design enables robust real-time control and adaptive local decision-making in complex, dynamic environments. By decoupling training, the VLM retains its generalization, while System 1 achieves interpretable and effective local navigation. DualVLN outperforms prior methods across all VLN benchmarks and real-world experiments demonstrate robust long-horizon planning and real-time adaptability in dynamic environments.

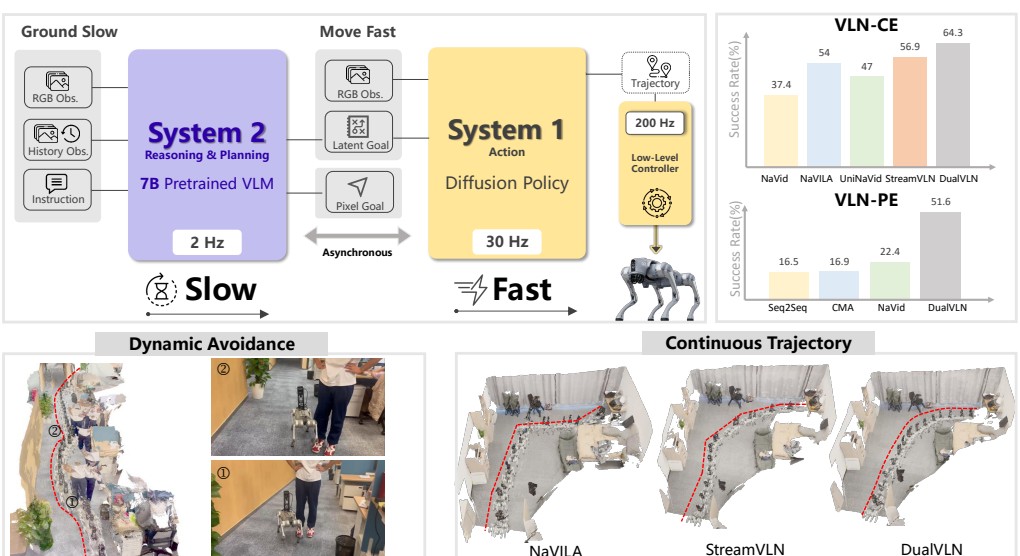

Figure 1: The proposed dual-system framework decouples high-level reasoning from low-level control. System 2 (slow, 2 Hz) uses a 7B pretrained VLM to generate pixel goal and latent goal, while System 1 (fast, 30 Hz) is a lightweight diffusion-based policy that converts the goals into smooth trajectories with high-frequency RGB inputs. The asynchronous inference enables continuous and smooth navigation process. DualVLN sets a new state-of-the-art on VLN-CE and VLN-PE, and shows strong generalization in real-world deployments.

# 1 INTRODUCTION

Vision-Language Navigation (VLN) is a critical task in robotics. A VLN system receives language instructions with visual observations as input and plans a trajectory toward the goal. Recently, this field has witnessed substantial progress, evolving from early benchmarks that focus on discrete goal planning Anderson et al. (2018); Ku et al. (2020), to continuous action space formulations Krantz et al. (2020b), and further to physically realistic simulations with locomotion controllers Wang et al. (2025b); Cheng et al. (2025). Meanwhile, large vision language models (VLMs) offer new potential for VLN, as their strong prior knowledge can be transferred through post-training to empower VLN systems with unprecedented generalization across diverse instructions and environments.

However, even in continuous VLN benchmarks, existing Vision-Language-Action (VLA) models Zhang et al. (2025a); Cheng et al. (2025); Zheng et al. (2024); Wei et al. (2025) largely adopt a tightly coupled end-to-end paradigm, mapping vision and language inputs directly to short-horizon discrete actions (e.g., move forward 0.25 m). Such design introduces critical limitations for real-world deployment. First, it produces fragmented and unnatural motions, leading to high execution latency since every step depends on frequent calls to large VLMs. Second, by entangling vision-language reasoning, global planning, and local control into a single pipeline, these models lack explicit coordination across hierarchical decision levels. Consequently, they fall short in meeting advanced requirements such as agile control and dynamic obstacle avoidance.

To overcome these limitations, we propose the first dual-system VLN foundation model DualVLN that explicitly bridges the reasoning strength of VLMs with the agility required for real-time control. DualVLN decouples the VLN pipeline into two complementary systems. System 2, a large foundation VLM, performs slow but robust reasoning and produces explicit intermediate pixel goals. System 1, a lightweight diffusion-based policy model, transforms the grounded targets into continuous traversable trajectories, enabling robust collision avoidance in dynamic scenarios. For a better coordination between System 1 and System 2, we connect the two systems through latent representations. After the System 2 is trained with the pixel goal grounding task, we freeze the weights of System 2. Then we introduce a set of learnable latent queries and optimize them via prompt tuning. These queries extract compact latent features and serve as implicit goals for System 1.

**Why decoupled sequential training?** Decoupling enables each system to specialize: System 2 can scale with large multi-source reasoning data, while System 1 needs only a few low-level goal reaching data. System 1 further benefits from additional high frequent RGB inputs and asynchronous inference to achieve higher control frequency in dynamic settings. Crucially, this separation preserves the VLM's generalization when adapting to downstream low-level planning.

**Why use both explicit pixel goal and implicit latent goal?** Relying solely on explicit 2D pixel goals as guidance for System 1 fails to fully exploit the rich hidden features of the VLM, resulting in a shallow connection between reasoning and local planning and reducing the dual-system to a modular pipeline. Learning explicit pixel goals enhances System 2's interpretability and generalization. Building upon this, implicit latent features further provide richer and more adaptive guidance for System 1, enabling it to automatically extract task-relevant representations from the heterogeneous information encoded in the VLM's hidden states.

Experimental results show that DualVLN consistently surpasses prior state-of-the-art methods on both VLN-CE Krantz et al. (2020b) and VLN-PE Wang et al. (2025b) benchmarks. Real-world evaluations demonstrate its robust long-horizon planning, real-time trajectory execution, and dynamic obstacle avoidance across multiple robot platforms and diverse scenarios. We also introduce the first Social-VLN benchmark to evaluate navigation models on social awareness and task recovery in dynamic environments, where humanoid agents are placed along task trajectories.

# 2 RELATED WORK

**Vision-Language-Action Model for Navigation.** Recent studies leverage multi-modal large models as pretrained backbones for navigation, aiming to use their inherent commonsense knowledge to enhance performance. A common approach formulates navigation actions as text, treating the task as next-token prediction within LLMs Zheng et al. (2024); Zhang et al. (2024; 2025a); Gao et al. (2025); Wei et al. (2025); Wang et al. (2025d). Others, such as RoboPoint Yuan et al. (2025)

and NaviMaster Luo et al. (2025), frame navigation as pixel grounding but still require additional modules for execution. End-to-end methods like UniVLA Bu et al. (2025) and TrackVLA Wang et al. (2025c) map VLM latent features directly to continuous trajectories, but their synchronized frameworks limit high-frequency decision-making in dynamic environments. While some recent dual-system architectures FigureAI (2025); Shi et al. (2025); Bu et al. (2024) explore slow-fast reasoning, they focus on tabletop tasks and do not address long-horizon planning or cross-building navigation. We propose the first asynchronous dual-system architecture supporting long-horizon instruction following, accurate planning, and navigation in unseen environments.

**Visual Navigation Policy Learning.** Visual navigation enables reaching explicit goals while performing real-time obstacle avoidance. Traditional modular approaches Fox et al. (1997); Kramer & Stachniss (2012); Karaman & Frazzoli (2011); Williams et al. (2015); Zhou et al. (2020) rely on explicit localization and mapping but suffer from compounding errors, latency, and extensive hyperparameter tuning. End-to-end learning-based methods have been proposed to address these issues: GNM Shah et al. (2023a), X-Nav Wang et al. (2025a), RING Eftekhar et al. (2024), and X-Mobility Liu et al. (2024) improve zero-shot generalization across embodiments, while iPlanner Yang et al. (2023), ViPlanner Roth et al. (2024), FDM Roth et al. (2025), and S2E He et al. (2025) focus on efficient training and sim-to-real transfer. Image-goal navigation has also been explored by SLING Wasserman et al. (2023), ViNT Shah et al. (2023b), NoMad Sridhar et al. (2024), and NaviDiffuser Zeng et al. (2025). Our System-1 is an RGB-only visual navigation policy conditioned on latent goals from VLMs.

## 3 METHOD

As illustrated in Figure 2, our framework employs a dual-system design that realizes a synergy between high-level reasoning and low-level action execution. System 2, a VLM-based planner, performs global planning by predicting mid-term waypoints in image pixel space, providing spatially grounded targets. System 1, a multi-modal goal-conditioned diffusion policy, generates continuous trajectories conditioned on current observations and asynchronous latent features from System 2, enabling robust, real-time control in complex environments.

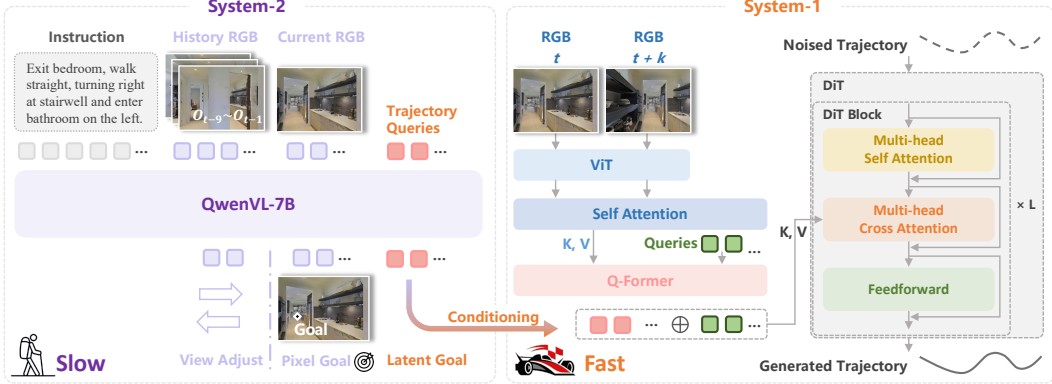

Figure 2: Overview of DualVLN. System 2 takes as input a sequence of egocentric images and the instruction to predict either view-adjustment actions or a 2D pixel coordinate within the image for the next navigation waypoint. System 1 then takes as input both the latent goal embeddings and high-frequency RGB inputs, then generates continuous trajectories for the robot to follow through a diffusion-based policy.

### 3.1 SYSTEM 2: VLM-BASED PIXEL-GOAL GROUNDING WITH SELF-DIRECTED VIEW ADJUSTMENT.

System 2 integrates high-level pixel-goal grounding with self-directed view adjustment in a iterative process. At each navigation step, the agent observes the current RGB frame and history, decides whether to adjust its view or output a pixel goal. This ensures that pixel goal predictions are based on informative perspectives, handling occlusions and challenging viewpoints.

**Farthest Pixel Goal Grounding.** We build our goal planning module upon Qwen-VL-2.5 Bai et al. (2025), a strong open-source vision-language model capable of spatial grounding in terms of image pixel coordinates. To adapt Qwen-VL-2.5 for vision-and-language navigation (VLN), we formulate high-level planning as a farthest pixel goal grounding problem. The model takes as input a sequence of egocentric RGB images along with the language instruction, and predicts 2-D coordinates within the image corresponding to the next preferred navigation waypoint. To generate training samples, we project the agent's 3-D trajectory onto the 2-D egocentric observations and measure the visibility from the agent's position. Specifically, before projecting the trajectory, we use the depth map together with the camera–point distance to identify which points fall within the visible region of the current view. Any trajectory point whose distance exceeds the corresponding depth value is treated as occluded and discarded. Based on this projection, we segment the original VLN-CE trajectories into pixel goal grounding samples.

**Self-Directed View Adjustment.** Projecting a 3-D trajectory onto 2-D pixel coordinates can be problematic. If the agent's viewpoint is too high, points on the floor may be occluded, while artificially lifting these points creates depth ambiguity, making it unclear where the actual target lies. Moreover, if the agent is facing the wrong direction, the next waypoint may lie outside the current field of view. Drawing inspiration from human navigation behavior—where people often look around and lower their gaze to the floor before selecting the next waypoint—System 2 autonomously decides when to scan the environment and adjust the camera angle, using discrete actions such as Turn Left/Right 15°, Look Up/Down 15°, actively seeking informative perspectives before predicting the next pixel goal.

### 3.2 System 1: A Diffusion Transformer Policy with MultiModal conditioning

**Latent Goal Representation.** After System 2 autoregressively generates the next pixel goal, the model naturally produces a context feature sequence $X$ encompassing the language instruction, historical images, current observations, view adjustment actions, and pixel goal information. We then append a set of learnable latent queries $Z$, which are randomly initialized and updated via prompt tuning. Processing the combined sequence $[X; Z]$ through VLM enables $Z$ to attend to and extract task-relevant semantic information from $X$. The resulting $Z'$ forms the intermediate latent goal representation, which conditions System 1 for precise, low-level trajectory generation.

**Multi-Modal Conditioning Diffusion Transformer.** System 1 is implemented as a diffusion transformer (DiT) that generates smooth trajectories (32 dense waypoints) for robots to follow with two sources of conditions: 1. Low-frequency trajectory latents $Z'$ from System 2. 2. High-frequency RGB inputs. Since the dual-system inference is performed asynchronously (Slow System 2, Fast System 1), the latent goal generated at time $t$ remains fixed. At time $t + k$, System 1 must still interpret this outdated latent goal to update the trajectory accurately, estimating the distance already traveled and adapting to dynamic changes.

To achieve this, System 1 encodes both the RGB features corresponding to the last frame from System 2 at time $t$ and the current observation at time $t + k$. Both images are first processed by a ViT encoder to extract high-dimensional visual features. These features are then fused across the two time steps using a self-attention module. To maintain fast inference, the fused features are further compressed using a Q-Former into a compact set of 32 tokens, which serve as high-frequency visual conditioning for the DiT.

*Flow Matching.* Given the ground truth trajectory waypoints $X_0$ and the two conditioning signals (trajectory latents $Z'$ and fused RGB tokens $F$), at each training step we first sample a diffusion timestep $u \sim \mathcal{U}(0, 1)$ and a noise vector $\epsilon \sim \mathcal{N}(0, I)$. The noisy trajectory is then defined as:

$$X_u = \alpha_u X_0 + \sigma_u \epsilon, \tag{1}$$

where $\alpha_u$ is a decreasing function of $u$ and $\sigma_u$ is an increasing function of $u$.

The diffusion transformer is trained to predict the velocity $\dot{X}_u$ of the trajectory at timestep $u$ conditioned on $Z'$ and $F$:

$$\hat{\dot{X}}_u = f_\theta(X_u, u, Z' \oplus F), \tag{2}$$

where $\oplus$ denotes concatenation, $f_\theta$ is the transformer network.

The training objective minimizes mean squared error between predicted velocity and true velocity:

$$\mathcal{L}_{\text{flow}} = \mathbb{E}_{u, X_0, \epsilon} \left[ \| \hat{\dot{X}}_u - \dot{X}_u \|_2^2 \right], \tag{3}$$

### 3.3 IMPLEMENTATION DETAILS.

For **System 2**, we follow the data recipe of StreamVLN Wei et al. (2025) and finetune QwenVL-2.5 (7B) for one epoch. Both the vision encoder and the LLM backbone are fully unfrozen during finetuning. For **System 1**, we introduce four learnable latent queries appended after the pixel goal prediction in System 2 to extract compact latent goal embeddings. The RGB encoder is implemented using the ViT backbone of DepthAnythingV2-Small. We adopt a compact Diffusion Transformer (DiT) design to ensure low-latency inference, with a hidden dimension of 384, 12 transformer layers, and 6 attention heads. The latent embedding size is linearly projected from 3584 to 768 before cross-attention with the DiT. More details can be found in Section A.

## 4 SOCIAL VISION-AND-LANGUAGE NAVIGATION BENCHMARK.

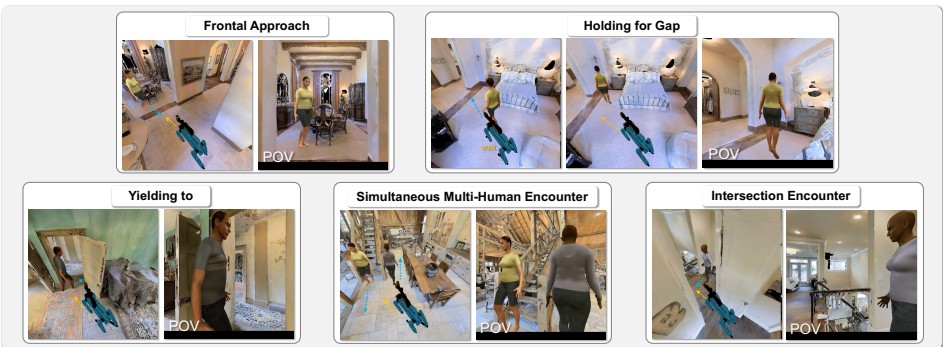

Figure 3: Typical robot-humanoid interactions that pose key challenges to the robot's human-aware obstacle avoidance capabilities, including not only situations with a single agent but also cases involving multiple humanoids simultaneously.

Despite recent progress in generalist navigation models, there lacks a benchmark designed to evaluate a model's ability to handle dynamic obstacles while executing task-oriented navigation. This ability is critical in human-centric environments, where robots must avoid collisions and can return to their original task trajectory after detouring. The VLN-CE benchmarks focus on static layouts, leaving a gap in assessing social awareness and trajectory recovery in dynamic settings.

**Benchmark Curation.** To highlight the importance of the dual-system design and further advance the development of generalist navigation agents, we extend the classic VLN evaluation from static setting to dynamic scenarios. We introduce Social-VLN, a new benchmark built upon R2R-CE Krantz et al. (2020b), by incorporating multiple dynamic agents into the simulation, modeled by humanoids provided in Habitat 3.0 Puig et al. (2023). Instead of letting humanoid agents wander randomly between arbitrary start and goal points, we place them strategically along the ground-truth VLN trajectory. Since most VLN tasks are relatively short-range, this targeted placement greatly increases the likelihood of nteractions, creating a challenging and realistic test of social VLN. Social-VLN enables a comprehensive assessment of socially-aware obstacle avoidance behaviors in diverse situations, as shown in Figure 3. We also carefully verify each episode to ensure agents do not block the path entirely, so that failures don't reflect just simple physical obstructions.

Building on the standard VLN metrics, we further introduce a **Human Collision Rate (HCR)** metric to explicitly quantify failures caused by unsafe interactions with dynamic pedestrians. Social-VLN evaluates not only task completion but also the agent's safety awareness in dynamic environments.

**Training Data Collection.** We also develop a pipeline for collecting dynamic obstacle-avoidance trajectories for training. In each training episode, a human detection sensor is setup to continuously monitored the egocentric view. When the human mask pixel ratio exceeded a predefined threshold, a modified A* algorithm was triggered to replan a collision-free trajectory. This process generated 763K social navigation episodes across 60 MP3D Chang et al. (2017) scenes, forming a foundational resource for training socially compliant agents.

# 5 EXPERIMENTS.

## 5.1 SIMULATION EXPERIMENTS.

Table 1: Comparison with state-of-the-art methods on VLN-CE R2R and RxR Val-Unseen split. ∗ indicates methods using the waypoint predictor from Hong et al. (2022).

| Method | Observation | | | | R2R Val-Unseen | | | | RxR Val-Unseen | | | |
|---|---|---|---|---|---|---|---|---|---|---|---|---|
| | Pano. | Odo. | Depth | S.RGB | NE↓ | OS↑ | SR↑ | SPL↑ | NE↓ | SR↑ | SPL↑ | nDTW↑ |
| HPN+DN∗ Krantz et al. (2021) | ✓ | ✓ | ✓ | | 6.31 | 40.0 | 36.0 | 34.0 | - | - | - | - |
| CMA∗ Hong et al. (2022) | ✓ | ✓ | ✓ | | 6.20 | 52.0 | 41.0 | 36.0 | 8.76 | 26.5 | 22.1 | 47.0 |
| GridMM∗ Wang et al. (2023a) | ✓ | ✓ | ✓ | | 5.11 | 61.0 | 49.0 | 41.0 | - | - | - | - |
| ETPNav∗ An et al. (2023) | ✓ | ✓ | ✓ | | 4.71 | 65.0 | 57.0 | 49.0 | 5.64 | 54.7 | 44.8 | 61.9 |
| ScaleVLN∗ Wang et al. (2023b) | ✓ | ✓ | ✓ | | 4.80 | – | 55.0 | 51.0 | - | - | - | - |
| InstructNav Long et al. (2024) | ✓ | ✓ | ✓ | ✓ | 6.89 | – | 31.0 | 24.0 | - | - | - | - |
| R2R-CMTP Chen et al. (2021) | ✓ | ✓ | ✓ | | 7.90 | 38.0 | 26.4 | 22.7 | - | - | - | - |
| LAW Raychaudhuri et al. (2021) | | ✓ | ✓ | ✓ | 6.83 | 44.0 | 35.0 | 31.0 | 10.90 | 8.0 | 8.0 | 38.0 |
| CM2 Georgakis et al. (2022) | | ✓ | ✓ | ✓ | 7.02 | 41.5 | 34.3 | 27.6 | - | - | - | - |
| WS-MGMap Chen et al. (2022) | | ✓ | ✓ | ✓ | 6.28 | 47.6 | 38.9 | 34.3 | - | - | - | - |
| ETPNav + FF Wang et al. (2024) | | ✓ | ✓ | ✓ | 5.95 | 55.8 | 44.9 | 30.4 | 8.79 | 25.5 | 18.1 | - |
| Seq2Seq Krantz et al. (2020b) | | | ✓ | ✓ | 7.77 | 37.0 | 25.0 | 22.0 | 12.10 | 13.9 | 11.9 | 30.8 |
| CMA Krantz et al. (2020b) | | | ✓ | ✓ | 7.37 | 40.0 | 32.0 | 30.0 | - | - | - | - |
| NaVid Zhang et al. (2024) | | | | ✓ | 5.47 | 49.1 | 37.4 | 35.9 | - | - | - | - |
| MapNav Zhang et al. (2025b) | | | | ✓ | 4.93 | 53.0 | 39.7 | 37.2 | - | - | - | - |
| NaVILA Cheng et al. (2025) | | | | ✓ | 5.22 | 62.5 | 54.0 | 49.0 | 6.77 | 49.3 | 44.0 | 58.8 |
| UniNaVid Zhang et al. (2025a) | | | | ✓ | 5.58 | 53.3 | 47.0 | 42.7 | 6.24 | 48.7 | 40.9 | - |
| StreamVLN Wei et al. (2025) | | | | ✓ | 4.98 | 64.2 | 56.9 | 51.9 | 6.22 | 52.9 | 46.0 | 61.9 |
| **DualVLN** | | | | ✓ | **4.05** | **70.7** | **64.3** | **58.5** | **4.58** | **61.4** | **51.8** | **70.0** |

**VLN-CE Benchmark & Metrics.** We first evaluate on the standard R2R-CE Anderson et al. (2018) and RxR-CE Ku et al. (2020) benchmarks, both built under the VLN-CE Krantz et al. (2020b) setting using the Habitat simulator. These benchmarks simulate realistic indoor navigation in Matterport3D environments, where agents follow natural language instructions under continuous control. All experiments are conducted on the validation unseen splits to assess generalization. Following prior work, we adopt standard VLN metrics: **Navigation Error (NE)**, measuring the final distance to the goal; **Success Rate (SR)**, the percentage of episodes where the agent stops within 3 meters of the goal; **Oracle Success Rate (OSR)**, where the closest point along the trajectory is considered; and **Success weighted by Path Length (SPL)**, which penalizes unnecessarily long paths.

**VLN-PE Benchmark & Metrics.** We further evaluate on VLN-PE Wang et al. (2025b), a physically realistic VLN platform that simulates robot dynamics and control errors in real-world deployment. We report results on the R2R dataset with the Humanoid Unitree H1 robot. In addition to the standard VLN metrics above, we further report **Trajectory Length (TL)**, **Fall Rate (FR)**, which measures the frequency of robot falls, and **Stuck Rate (StR)**, the occurrences where the agent is unable to move. These metrics collectively provide a comprehensive assessment of both the effectiveness and robustness of the system in continuous and physically realistic navigation scenarios.

**Result Analysis.** As shown in Table 1, we compare DualVLN under the VLN-CE evaluation against three representative categories of baselines: (1) Multi-sensor methods that incorporate panoramic RGB, odometry, and depth (e.g., HPN+DN, CMA, GridMM, ETPNav); (2) VLM-free methods trained on single first-person RGB and depth (e.g., CM2, LAW, WS-MGMap); (3) Video-LLM based methods relying solely on single-view RGB (e.g., NaVid, MapNav, NaVILA, UniNaVid, StreamVLN). With only first-person RGB inputs, DualVLN achieves substantial gains over all prior RGB-based approaches, highlighting the strength of our dual-system design.

Table 2 reports VLN-PE results with the physical locomotion controller. Baselines include Seq2Seq Krantz et al. (2020b), CMA Krantz et al. (2020b), RDP Wang et al. (2025b), and NaVid Zhang et al. (2024). Seq2Seq predicts actions from RGBD inputs with a recurrent policy, while CMA adds cross-modal attention with instructions. RDP introduces a Transformer diffusion decoder for continuous displacements, and NaVid leverages video-based LLMs for improved generalization without depth or odometry. Despite not being fine-tuned on VLN-PE trajectories, DualVLN surpasses all baselines, including those trained on VLN-PE and VLM-based methods.

Table 2: Evaluation Metrics on VLN-PE benchmark with physical locomotion controller. +: model is first trained on Habitat and fine-tuned on VLN-PE. †: model is trained with data augmentation.

| Method | R2R Validation Seen | | | | | | R2R Validation Unseen | | | | | |
|---|---|---|---|---|---|---|---|---|---|---|---|---|
| | TL↓ | NE↓ | FR↓ | StR↓ | OS↑ | SR↑ | SPL↑ | TL↓ | NE↓ | FR↓ | StR↓ | OS↑ | SR↑ | SPL↑ |
| Train on VLN-PE | | | | | | | | | | | | |
| CMA | 11.13 | 7.59 | 23.71 | 3.19 | 34.94 | 21.58 | 16.10 | 11.16 | 7.98 | 22.64 | 3.27 | 33.11 | 19.15 | 14.05 |
| CMA+ | 8.86 | 7.14 | 23.56 | 3.50 | 36.17 | 25.84 | 21.75 | 8.70 | 7.26 | 21.75 | 3.27 | 31.40 | 22.12 | 18.65 |
| RDP | 13.26 | 6.76 | 27.51 | 1.82 | 38.60 | 25.08 | 17.07 | 12.70 | 6.72 | 24.57 | 3.11 | 36.9 | 25.24 | 17.73 |
| Zero-shot Transfer Evaluation from VLN-CE | | | | | | | | | | | | |
| Seq2Seq† | 7.80 | 7.62 | 20.21 | 3.04 | 19.30 | 15.20 | 12.79 | 7.73 | 7.18 | 18.04 | 3.04 | 22.42 | 16.48 | 14.11 |
| CMA† | **6.62** | 7.37 | 20.06 | 3.95 | 18.54 | 16.11 | 14.64 | **6.58** | 7.09 | 17.07 | 3.79 | 20.86 | 16.93 | 15.24 |
| NaVid | 7.54 | 6.20 | **11.25** | **0.46** | 24.32 | 21.58 | 17.45 | 7.12 | 5.94 | **8.61** | **0.45** | 27.32 | 22.42 | 18.58 |
| **DualVLN** | 10.65 | **4.13** | 17.78 | 1.82 | **62.31** | **58.97** | **47.78** | 10.09 | **4.66** | 12.32 | 2.23 | **55.9** | **51.60** | **42.49** |

**Social-VLN Experiment.** We evaluate DualVLN and StreamVLN on the Social-VLN benchmark. StreamVLN is selected as the baseline due to its low action latency, which allows it to react to dynamic obstacles to some extent. As shown in Table 3, both methods experience substantial performance drops — e.g., the success rate of DualVLN decreases by about 27% and that of StreamVLN by 26% compared to their results on standard VLN tasks — highlighting the increased difficulty of Social-VLN setting. DualVLN achieves better task completion performance with obstacle avoidance than StreamVLN. Nevertheless, there remains considerable room for improvement on this task. We show some qualitative results in Figure 4.

Table 3: Comparison of DualVLN and StreamVLN on standard R2R VLN and Social-VLN.

| Method | R2R Val-Unseen (VLN) | | | | R2R Val-Unseen (Social-VLN) | | | | |
|---|---|---|---|---|---|---|---|---|---|
| | NE↓ | OS↑ | SR↑ | SPL↑ | NE↓ | OS↑ | SR↑ | SPL↑ | HCR↓ |
| StreamVLN | 4.98 | 64.2 | 56.9 | 51.9 | 6.50 | 36.3 | 31.4 | 29.1 | 36.4 |
| DualVLN | 4.05 | 70.7 | 64.3 | 58.5 | **5.97** | **41.0** | **37.2** | **35.8** | **35.4** |

## 5.2 REAL-WORLD CROSS-EMBODIMENT EXPERIMENTS

**Experimental Setup.** We perform real-world experiments on wheeled (Turtlebot4), quadruped (Unitree Go2) and humanoid (Unitree G1) robots. All are equipped with Intel RealSense D455 cameras mounted at varying heights and angled downward by 15°. The full model runs on a remote server with an RTX 4090 GPU, occupying 20GB memory. Given a VLN instruction, the robot streams synchronized RGB-D images to a remote server for asynchronous inference with the dual-system model. The server outputs trajectories or discrete view adjustment actions, transformed into world coordinates via odometry and tracked with an MPC controller. System 2 exploits KV-cache reuse to reduce trajectory token inference from 1.1s to 0.7s, while System 1 generates 32 trajectories in parallel within 0.03s using TensorRT. This asynchronous pipeline ensures a fresh trajectory is always available, yielding smooth, near real-time navigation.

**Quantitative Analysis.** To quantitatively assess DualVLN's robustness and generalization in real-world settings, we benchmarked it against CMA Krantz et al. (2020a), and VLM-based methods

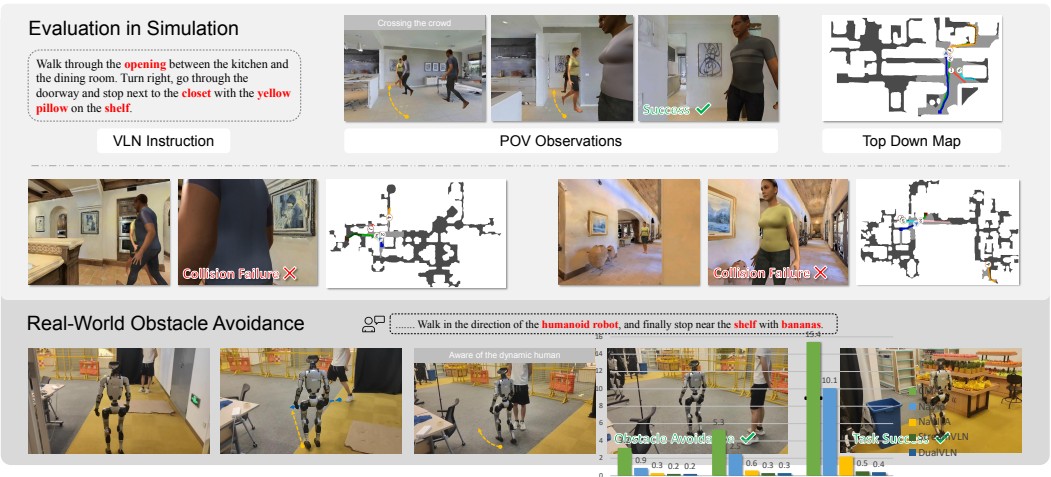

Figure 4: Qualitative Results of Social-VLN Experiments.

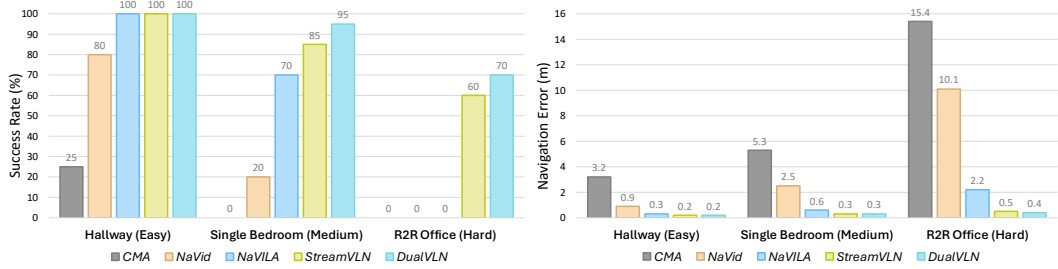

Figure 5: Evaluation Metrics of Real-World Experiments.

NaVid Zhang et al. (2024), NaVILA Cheng et al. (2025), StreamVLN Wei et al. (2025) which outputs discrete actions. Evaluations were conducted across hallway (easy), bedroom (medium), and office (hard, room-to-room) scenarios, with 20 trials per scenario per model. Performance was measured using Success Rate (SR) and Navigation Error (NE). Among VLM-based baselines shown in Figure 6, NaVid struggles with complex task, NaVILA handles long-horizon tasks but often misses the final goal in office scenarios. StreamVLN avoids obstacles in some cases but sacrifices task completion. Our dual-system DualVLN consistently achieves high SR and low NE across both static and dynamic scenarios.

**Qualitative Analysis.** Please refer to the supplement video. We evaluate with diverse real-world scenarios, including office, canteen, street, and convenience store, in a zero-shot setting without scene-specific finetuning. DualVLN can select correct pixel goals and produces safe trajectories in cluttered environments, plans smooth paths passing all desired landmarks, and handles staircases and dynamic pedestrians. Moreover, the dual-system performs robustly across different robot platforms despite variations in camera height, vibration, and tracking.

## 5.3 ABLATION STUDY.

**Impact of Explicit Pixel and Latent Goal.** To assess the role of different goal representations in conditioning System-1 of DualVLN, we perform a series of ablation studies as shown in Figure 7. We first consider an alternative design without sequential training, where System 1 is trained end-to-end jointly with System 2 and does not rely on explicit pixel goals. In this setup (*w/o Sys.2 Train*), we observe that the diffusion policy converges significantly more slowly, and System 2's generalization ability deteriorates. This confirms that decoupled training with intermediate pixel goals is crucial for both efficient learning and preserving the reasoning strength of the VLM.

Secondly, during System 1 training stage, we remove the explicit pixel-goal text from the context sequence $X$ before appending the latent queries $Z$. In this case (*w/o Pixel Goal*), the latent goal features cannot attend to explicit pixel-goal information. This leads to a clear performance drop,

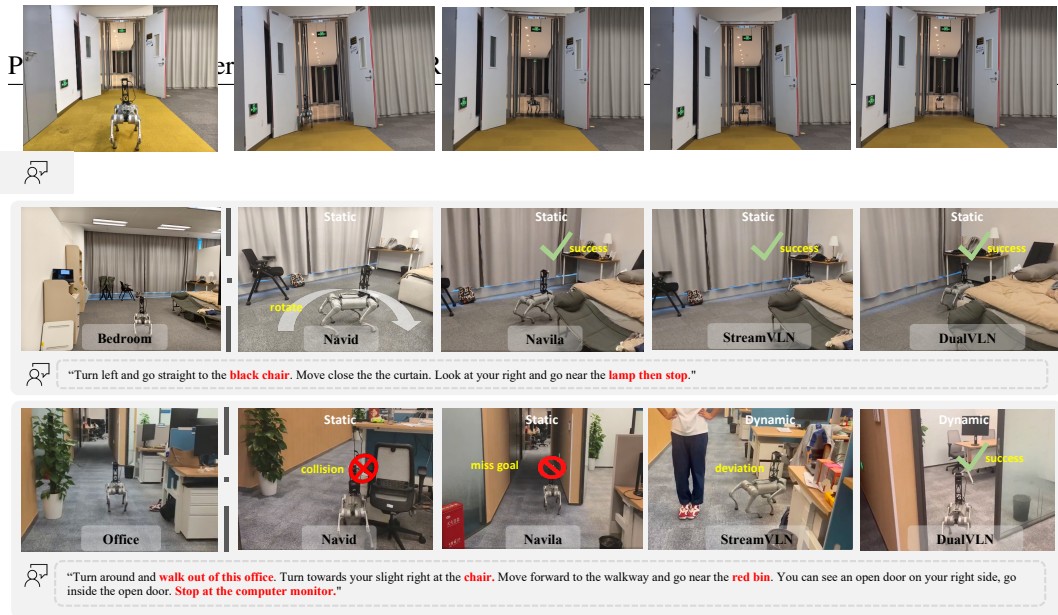

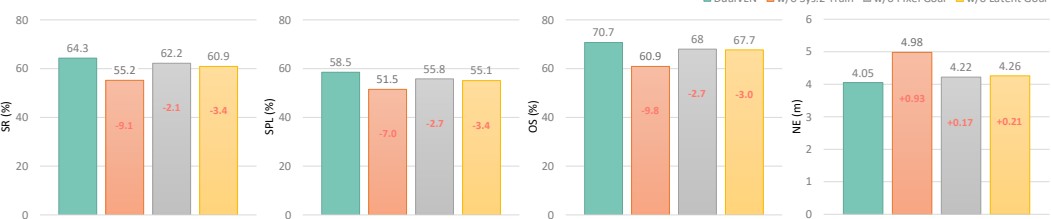

Figure 6: Real-World Performance Analysis of VLM-based methods.

Figure 7: Ablation Study on the role of different goal representations in conditioning System-1. *w/o Sys.2 Train* means training System 1 & 2 jointly **in one-stage** without explicit intermediate pixel goals. *w/o Pixel Goal* means removing the pixel-goal text before appending the latent queries. *w/o Latent Goal* means using the frozen VLM hidden states of the generated pixel goal.

confirming that explicit pixel goals provide valuable guidance for the diffusion policy while also enhancing interpretability and generalization.

Finally, we consider a variant where only the last-layer VLM hidden states of the pixel-goal text are used as the conditioning signal for System 1. This setup (*w/o Latent Goal*) yields weaker performance. The reason is that, without latent goal queries, System 1 is restricted to passively consuming fixed VLM features rather than learning which hidden states should serve as conditioning. This limits the adaptive information flow from System 2 to System 1.

Table 4: Ablation study of different local planner on VLN-PE benchmark with flash controller.

| Local Planner | R2R Validation Seen | | | | | R2R Validation Unseen | | | | |
|---|---|---|---|---|---|---|---|---|---|---|
| | TL↓ | NE↓ | OS↑ | SR↑ | SPL↑ | TL↓ | NE↓ | OS↑ | SR↑ | SPL↑ |
| iPlanner | **10.9** | 4.08 | 66.26 | 58.66 | 49.43 | 9.58 | 4.91 | 55.53 | 47.07 | 41.09 |
| NavDP | 11.68 | 3.75 | 76.44 | 66.11 | 56.26 | 10.18 | 4.22 | 67.33 | 58.72 | 50.98 |
| System 1 | 11.26 | **3.15** | **78.42** | **73.25** | **64.00** | 10.08 | **3.90** | **69.93** | **63.62** | **56.49** |

**System 1 vs. SOTA Point-Goal Navigation Policies.** To validate the advantage of our dual-system joint training framework, we remove the latent goal and convert the explicit pixel goal into a point goal using additional depth information. We then integrate state-of-the-art point-goal navigation policies (e.g., iPlanner Yang et al. (2023) and NavDP Cai et al. (2025)) to replace System 1 as the local planner. The results shown in Table 4 demonstrate that, even with oracle depth, such a modular pipeline performs worse than our dual-system approach. We attribute this performance gap to two key factors: (1) the trajectory distribution gap between those produced by point-goal planners and System 2's training data leads to degraded pixel-goal prediction; and (2) System 1 exhibits strong vision-based obstacle-avoidance behavior which makes it robust to small pixel-goal deviations in the correct direction, maintaining accurate, obstacle-aware trajectories, but not to large or semantically incorrect goals (see Figure 8). In contrast, point-goal is highly sensitive to even minor pixel errors by directly projecting the pixel goal into a world-coordinate point.

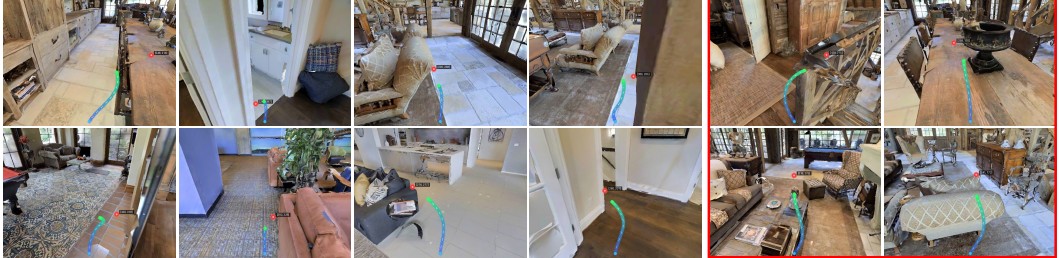

Figure 8: System 1 is robust to pixel-goal regression errors that still indicates the correct direction but may place the goal near or on an obstacle. But this robustness does not extend to large or semantically incorrect pixel goals especially when the agent is close to the obstacles.

**Data Scaling Analysis of System-1.** For the data scaling of System 2, we observe a similar trend with Navila Cheng et al. (2025) and StreamVLN Wei et al. (2025): more diverse data consistently improves performance, reflecting the data-hungry nature of VLM. In contrast, as shown in Figure 9, System 1 exhibits a different scaling behavior. Since it is designed to be lightweight and fast, and the task itself is relatively simple, even using only 1% of the trajectories collected for System 2 already yields competitive performance. Scaling to 10% leads to near-saturation. Further increasing the data scale to 50% does not bring significant additional gains, indicating that the performance upper bound of System 2 has been reached.

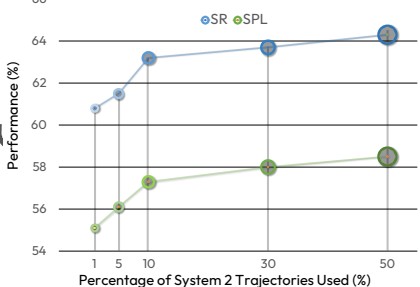

Figure 9: Data Scaling Results of Sys 1.

**Consistency between Pixel Goal and Trajectory.** To verify that trajectory prediction are strongly guided by the pixel goal, we analyze their consistency by projecting the predicted trajectory points onto the image plane. Using 1000 random samples from DualVLN models with different success rates on the VLN-CE benchmark, we compute two metrics: the pixel distance between the projected trajectory and the pixel goal, and their average angular deviation. As shown in Figure 10, most points are concentrated in the lower-left region of the plot, indicating that the trajectories are oriented toward the pixel goal and reach areas near the pixel goal.

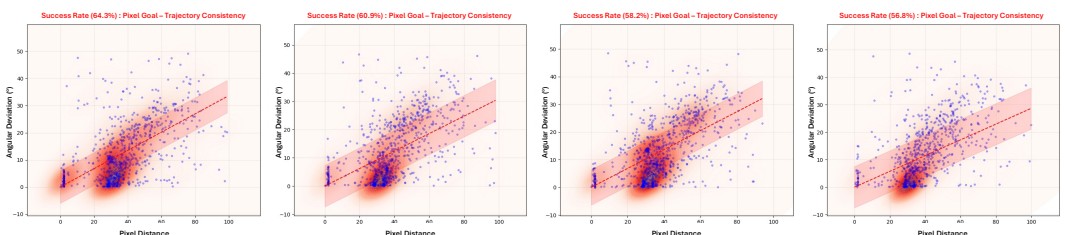

Figure 10: Correlation between Predicted Pixel Goal and Trajectory

## 6 CONCLUSION

In this work, we presented DualVLN, a dual-system vision-language navigation foundation model that decouples high-level semantic grounding from low-level action execution. By combining explicit pixel-grounded waypoints with implicit latent goal representations, DualVLN enables more robust, efficient, and generalizable navigation compared to existing end-to-end and modular approaches. Our dual-system joint training framework bridges the gap between semantic reasoning and motion control, producing smoother trajectories and demonstrating strong performance across diverse environments and tasks. We believe DualVLN offers a flexible and scalable foundation for future embodied navigation systems, and we hope it inspires further research toward more general-purpose, multimodal, and real-world-ready embodied agents.

# 7 CONTRIBUTIONS AND ACKNOWLEDGMENTS

**Model**: Meng Wei, Xiqian Yu, Jiaqi Peng, Wenzhe Cai, Delin Feng, Chenming Zhu

**VLN Data Curation**: Chenyang Wan, Meng Wei

**Simulation & Benchmarking**: Meng Wei, Chenyang Wan, Delin Feng, Yuqiang Yang, Wenzhe Cai

**Real-world Deployment**: Yuqiang Yang, Meng Wei, Jiaqi Peng

**Advising**: Tai Wang, Jiangmiao Pang, Xihui Liu

**Acknowledgments**: The research work described in this paper was conducted in the JC STEM Lab of Autonomous Intelligent Systems funded by The Hong Kong Jockey Club Charities Trust. This research is supported by Shanghai Artificial Intelligence Laboratory. This work offers a comprehensive elaboration and introduces some model improvements for the Dual-System Vision-Language Navigation (VLN) Foundation Model component within the InternVLA-N1 framework. We would like to extend our sincere gratitude to all collaborators for their contributions to InternVLA-N1 InternNav-Team (2025) and the InternNav InternNav-Contributors (2025) codebase, encompassing data collection, model development, simulation, benchmarking, real-robot deployment and open-source efforts: `Peizhou Cao, Yilun Chen, Zeyu He, Yifei Huang, Wensi Huang, Hengjie Li, Yu Liu, Dahua Lin, Jingli Lin, Yilin Long, Xiaohan Mao, Yu Qiao, Jiawei Qiu, Yuan Shen, Yukai Wang, Hanqing Wang, Liuyi Wang, Xueyuan Wei, Chao Wu, Zhenyu Yang, Jia Zeng, Yiming Zeng, Siqi Zhang, Jingjing Zhang, Shenghan Zhang, Shi Zhang, Yuchang Zhang, Hui Zhao, Bowen Zhou, Yuanzhen Zhou, Haoyi Zhu, Shaohao Zhu` *(listed in alphabetical order by their last names).*

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

# SUPPLEMENTARY MATERIAL

## A   DATA PREPARATION AND TRAINING DETAILS

### A.1   SYSTEM 2: QWENVL2.5

System 2 is trained to predict three types of outputs based on the agent's observations and instructions: **self-directed view adjustments**, **pixel-goal grounding**, and **STOP** actions.

**Discrete Ground-Truth Action Set**  `0:  STOP, 1:  Move forward 25 cm, 2:`
`Turn left 15°, 3:  Turn right 15°`

**User Prompt Template**

```
User: You are an autonomous navigation assistant. Your task is
<instruction>. Where should you go next to stay on track? Please
output the next waypoint's coordinates in the image.
Please output STOP when you have successfully completed the task.
These are your historical observations: <history>.
```

**Self-Directed View Adjustment**   When the future trajectory cannot be projected onto the current observation (e.g., at the start of consecutive turn actions), the model predicts upcoming turn actions instead of pixel goals. Maximum of 4 consecutive turn actions are predicted per chunk.

Example Converstion:

```
User: You are an autonomous navigation assistant. ... <history>.
      Your current observation is <image>
Assistant: → → → →  (corresponding to Turn right 60 degrees)
```

**Pixel-Goal Grounding**   When at least one future waypoint is visible in the current observation, the model is supervised to predict the **farthest successfully projected waypoint** as the pixel goal.

Example Converstion:

```
User: You are an autonomous navigation assistant. ... <history>.
      Your current observation is <image>
Assistant: ↓ (indicating that the next pixel goal is in view )
User: Your current observation is <image>  (it's optional to take
      a looking down action)
Assistant: 234 447  # Pixel goal text
```

**STOP**   The last step is supervised to output `STOP`.

Example Converstion:

```
User: You are an autonomous navigation assistant. ... <history>.
      Your current observation is <image>
Assistant: STOP
```

**Training**   In Stage 1 training, QwenVL is fully finetuned to **autonomously** produce either turn-action sequences or coordinate text or stop depending on the vision and language context.

We use the AdamW optimizer with an initial learning rate of $2e-5$ for full finetuning. Training is conducted with a batch size of 128 conversation samples and runs for a total of 14,000 steps.

A.2 System 1: Diffusion-Based Trajectory Policy

**Smooth and Resample Discrete Action Waypoints**   Discrete action waypoints are converted into 32 smooth fixed-interval trajectory waypoints via interpolation.

**Extracting Latent Representations from QwenVL.**   To provide informative latent representations of the pixel goal for System 1, we append four special tokens `<TRAJ>` to the end of the text sequence in the pixel-goal grounding data. For example:

```
User: <observation t>
Assistant: ↓
User: <look-down observation t>
Assistant: (234, 447)
<TRAJ><TRAJ><TRAJ><TRAJ>
```

These latent queries are inserted into the QwenVL embeddings before the forward pass:

```
inputs_embeds = QwenVLModel.embed_tokens(input_ids)
traj_token_idx = (input_ids == TRAJ_TOKEN_INDEX)
inputs_embeds[traj_token_idx] = latent_queries
hidden_states = QwenVLModel.forward(inputs_embeds)
pixel_goal_latents = hidden_states[-1][:, -N_QUERY:, :]
```

The extracted latent representations are then fed into the DiT-based diffusion policy to generate smooth, obstacle-aware trajectories:

```
x = Trajectory_Encoder(gt_rel_pose_list)  # relative poses
noise_pred = DiT(x, timestep, pixel_goal_latents)
```

**Training**   Only pixel-goal grounding samples are used for trajectory supervision. QwenVL is frozen; only the following modules are trained:

1. **Latent Queries**: Learnable embeddings to extract latent goal representations from frozen QwenVL.
2. **DiT-Based Diffusion Policy**: Generates smooth, obstacle-aware world-coordinate trajectories conditioned on latent goal representation.

Stage 2 end-to-end trains the latent representation and the diffusion policy to predict obstacle-aware trajectories in a parameter-efficient manner. The progressive two-stage training of DualVLN ensures both generalized pixel goal grounding and robust trajectory execution.

We use the AdamW optimizer with an initial learning rate of $1e-4$, a batch size of 128 trajectory sample, and train for a total of 15,000 steps.

B Attention Map Analysis for Pixel-Goal Grounding

To gain more insights into how System 2 (QwenVL) grounds pixel goals, we visualize its attention maps over both the language instructions and the visual inputs, including historical video frames and the current observation. In Figures 11, Attention maps from different transformer layers are presented to illustrate which aspects of the multi-modal context the model attends to when predicting the next pixel goal.

We observe that in the shallower transformer layers, the model primarily attends to general contextual and spatial cues such as objects, scene layouts, and directional clues in both language and visual tokens. As the transformer layers going deeper, the attention begins to increasingly concentrates on the specific target pixel goal region. This indicates a progressive refinement from broad scene and semantic context understanding toward precise, goal-directed pixel localization.

We also notice that in the deepest transformer layers, the model assigns significant attention weight to the STOP token when predicting the next action. This demonstrates that the model integrates cues from both visual and language inputs across all layers to make a final decision on task completion.

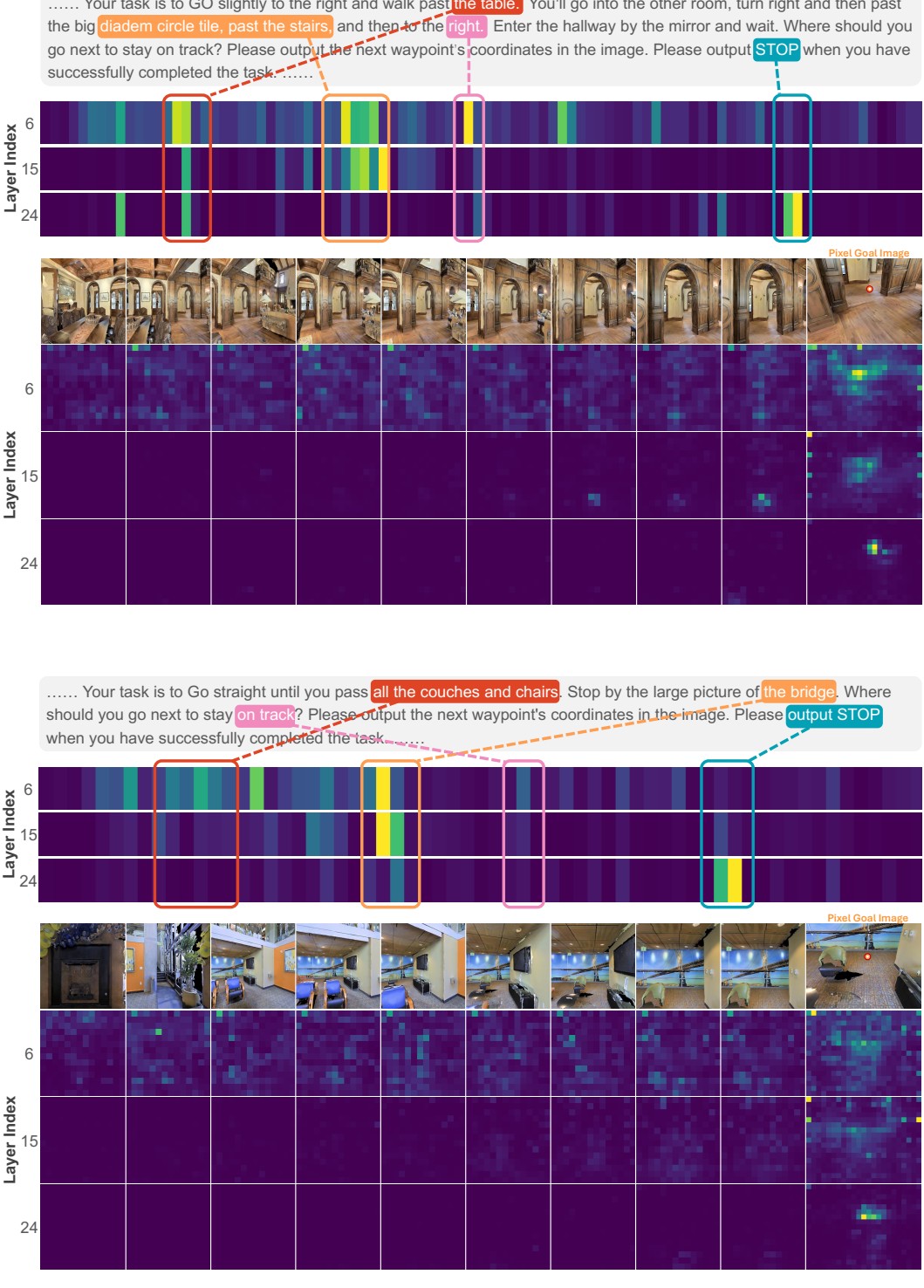

Figure 11: Visualization of attention maps when predicting the pixel goal.

