# OpenReview forum: "Ground Slow, Move Fast: A Dual-System Foundation Model for Generalizable Vision-Language Navigation"
_ICLR.cc/2026/Conference — ICLR 2026 Poster_

### Official Review · Reviewer_GZFu · 2025-10-25

**Soundness:** 3
**Presentation:** 4
**Contribution:** 3
**Rating:** 8
**Confidence:** 4

**Summary:**

The paper presents DualVLN, a dual-system foundation model for vision-language navigation (VLN). The proposed architecture decouples slow, global reasoning from fast, local control. System 2, built upon QwenVL-2.5 (7B), performs low-frequency, image-grounded reasoning to predict mid-term waypoints via pixel goal grounding and latent goal representation. System 1, a lightweight diffusion transformer policy, generates smooth, real-time trajectories conditioned on these goals and high-frequency RGB inputs.
This asynchronous slow-fast design enables both high-level reasoning and agile local control. DualVLN achieves state-of-the-art results on VLN-CE and VLN-PE benchmarks and shows strong real-world generalization across multiple robot embodiments. The authors further introduce Social-VLN, a new benchmark evaluating social awareness and dynamic obstacle avoidance.

**Strengths:**

1. Novel dual-system design that elegantly separates global reasoning and real-time control, addressing key challenges of latency and fragmented motion in VLN.
2. Pixel goal grounding is a well-motivated and technically sound intermediate representation, providing interpretable supervision and improving semantic grounding.
3. Strong empirical results: DualVLN consistently outperforms existing VLM-based and diffusion-policy baselines on VLN-CE, VLN-PE, and real-robot experiments.
4. Comprehensive evaluation, including the new Social-VLN benchmark, ablation studies, and cross-embodiment real-world tests, demonstrating robustness and scalability.

**Weaknesses:**

1. The paper frames DualVLN as a reasoning–acting foundation model, but the reasoning component is mostly implicit. System 2 performs spatial grounding and waypoint prediction rather than explicit multi-step or interpretable reasoning, making the “foundation-level reasoning” claim somewhat overstated.
2. The main novelty lies in the dual-system and asynchronous architecture, which is well-engineered but incremental. Similar slow–fast paradigms have appeared in StreamVLN, Helix, and Hi-Robot; the paper could clarify what new learning principle is introduced beyond system integration.
3. The Social-VLN benchmark is valuable yet briefly analyzed. Deeper quantitative or behavioral analysis would better support its claimed generalization and practical significance.

**Questions:**

Could the authors provide qualitative analysis or visualizations showing how System 2 grounds the pixel goal (e.g., attention maps or reasoning traces)?

---

> ### Author Response · Authors · 2025-11-21
> **Response to Reviewer GZFu**
>
> 1. ### ❓W1: System 2 performs spatial grounding and waypoint prediction rather than explicit multi-step or interpretable reasoning.
>   - The dual-system framework ultimately aims to predict world-coordinate waypoints from RGB input alone, which is highly challenging for 2D VLMs. By the dual-system design and predicting 2D pixel goals first, the model can establish preliminary goal-position-based reasoning, effectively decomposing the task and reducing its difficulty. This step is crucial before incorporating high-level language-based reasoning.
>   - In future work, we anticipate extending this framework to include more explicit reasoning at key steps—especilally the very human-like self-view adjustment ability — to further improve interpretability and performance.
> 2.  ### ❓W2: Similar slow–fast paradigms have appeared in StreamVLN, Helix, and Hi-Robot; the paper could clarify what new learning principle is introduced beyond system integration.
>   - StreamVLN is a single-system approach: although it leverages a sliding-window KV cache to speed up inference in the single-system VLA framework, each action prediction still requires a forward pass through a 7B VLM. In contrast, in our dual-system framework, System 1 has a much smaller footprint (only 100M parameters) and can output multiple actions at once. Even when considering the same action-step intervals, our framework achieves an inference frequency 1.6× higher than StreamVLN, demonstrating a significant efficiency improvement.
>   - Hi-Robot is not a slow–fast dual-system VLA. Its high-level planner interprets open-ended instructions to produce language commands (e.g., decomposing “Can you get me some chips, Oreos, and drinks?” into steps like: 1. pick up orange chip bag, 2. put chip bag in basket, 3. pick up Oreo, 4. put Oreo in basket). The actual action execution, however, entirely relies on a powerful VLA model such as Pi0 (>3B parameters). Importantly, there is no parallel operation or mutual feedback between the high-level planner and the VLA model, unlike a true slow–fast dual-system design.
>   -  While Helix is also a dual-system VLA, it trains both systems end-to-end using latent features without explicit intermediate goals. In this setup, System 2 (the VLM) is updated via gradients backpropagated from System 1 which outputs out-of-domain low-level actions. This end-to-end fine-tuning of the VLM can introduce catastrophic forgetting, reducing the VLM’s generalization ability and widening the sim-to-real gap. We also experimented with omitting pixel-goal supervision for Stage-1 System 2 training (Fig. 6, “w/o Sys.2 Train”) and observed performance drops. VLN is a long-horizon task that requires System 2 to perform stronger reasoning and maintain extended scene-level memory, which is significantly more demanding than manipulation tasks.
>
> 3. ### ❓W3: The Social-VLN benchmark is valuable yet briefly analyzed.
> - We thank the reviewer for this insightful comment. We have updated Section 4 with additional analysis.We will also release the benchmark code, and data to support future research and facilitate deeper community exploration.
> 4. ### ❓W4: Qualitative analysis or visualizations showing how System 2 grounds the pixel goal?
> - We appreciate your valuable suggestion about this qualitative analysis that makes things more explainable.
> In the revised paper, we have added qualitative analyses and visualizations of how System 2 grounds the pixel goal, including QwenVL attention-map analyses. Please refer to Appendix Section B (after page 14) for the detailed visualizations and discussion.

---

> > ### Comment · Reviewer_GZFu · 2025-11-24
> >
> > Thanks for your response, my concerns are well addressed and I keep my postive rating.

---

> > > ### Author Response · Authors · 2025-11-27
> > >
> > > Thanks again for the valuable suggestions.

---

### Official Review · Reviewer_EsMj · 2025-10-28

**Soundness:** 3
**Presentation:** 1
**Contribution:** 3
**Rating:** 4
**Confidence:** 4

**Summary:**

This paper proposes a dual-system framework (DualVLN) for VLN, consisting of System 2 and System 1. The authors mention that this framework can leverage the strong reasoning capabilities of VLMs and the fast inference speed of diffusion policies. While the implementation is described comprehensively, some concepts lack clarity. DualVLN is evaluated on multiple benchmarks, including VLN-CE, R2R, RxR, and the proposed Social-VLN benchmark. Results from both benchmark tests and real-world experiments demonstrate the effectiveness of DualVLN.

**Strengths:**

- The dual-system design appears well-designed, and the results demonstrate that DualVLN achieves both a high success rate and fast execution speed.
- DualVLN has been extensively evaluated in both synthetic and real-world environments.
- The proposed Social-VLN benchmark could benefit the research community by providing more realistic VLN environments.

**Weaknesses:**

The major weakness of this paper is that many statements and descriptions are unclear or confusing:
 - L173 mentions "self-directed view adjustment capabilities," but no details are provided on how this capability is implemented. It remains unclear how the VLA models are trained to possess this ability or how it is utilized during inference.
- L83 states that the training of System 1 and System 2 is decoupled. This raises the question of how the "Trajectories Query" in Figure 2 is learned within QwenVL. Specifically, what supervision signal is used for the latent trajectories?
- L425 references a "Table X," which appears to be missing. This table seems crucial for evaluating the effectiveness of pixel goal prediction.
-  The format of the pixel goal is not specified. Is it represented using special tokens or simple coordinate text?

**Questions:**

- Please refer to the 'Weaknesses' section for detailed concerns.
- Providing visualizations of the pixel goal in both synthetic and real-world environments would help readers better understand this concept.
- The rationale for using a pixel goal instead of directly predicting waypoints relative to the robot. Waypoints may seem more straightforward.

If the authors can address these concerns, I would be happy to raise my score.

---

> ### Author Response · Authors · 2025-11-21
> **Response to Reviewer EsMj**
>
> In the revised version, We have added new supplementary materials, including **an expanded appendix (after page 14)** and **demonstration videos**, to further clarify our method.
> 1. ### ❓W1&Q2: How the VLA models are trained to possess "self-directed view adjustment capabilities". Providing visualizations of the pixel goal in both synthetic and real-world environments.
>   - Please first refer to the General Response.
>   - We show visualizations of the interplay of view-adjustment actions, pixel-goal grounding, and trajectory prediction in the new supplemental videos.
> 2. ### ❓W2: How the "Trajectories Query" in Figure 2 is learned within QwenVL?  What supervision signal is used for the latent trajectories?
>   - We append 4 special tokens `<TRAJ>` to the end of the text sequence of the pixel goal grounding data sample:
>     ```text
>     User: <observation t>
>     Assistant: ↓
>     User: <look-down observation t>
>     Assistant: (234, 447)
>     <TRAJ><TRAJ><TRAJ><TRAJ>
>     ```
>     Pseudo-code:
>     ```python
>     # Embed input tokens
>     inputs_embeds = QwenVLModel.embed_tokens(input_ids)
>
>     # Find TRAJ token indices
>     traj_token_idx = (input_ids == TRAJ_TOKEN_INDEX)
>
>     # Replace embeddings with latent queries
>     inputs_embeds[traj_token_idx] = latent_queries
>
>     # Forward pass through QwenVL
>     hidden_states = QwenVLModel.forward(
>         inputs_embeds,
>         position_ids=position_ids,
>         attention_mask=attention_mask
>     )
>
>     # Extract latent representations corresponding to TRAJ tokens
>     pixel_goal_latents = hidden_states[-1][:, -N_QUERY:, :]
>
>     x = Trajectory_Encoder(gt_rel_pose_list) # relative poses of 32 waypoints
>     # Use the latent goal for DiT-based diffusion policy
>     noise_pred = DiT(x, timestep, pixel_goal_latents)
>     ```
>   - During this training stage, only the latent query and the DiT modules are updated. The latent query is supervised by System 1’s target, which predicts 32 smoothed future trajectory waypoints. Consequently, the latent query both consolidates and transmits the abstract goals and contexts encoded in the VLM’s hidden states, while preserving QwenVL’s original embedding space and preventing catastrophic forgetting during end-to-end fine-tuning.
>
> 3. ### ❓W3: L425 references a "Table X," which appears to be missing.
>   - Thanks for pointing this out. We have fixed the misreference in the revised manuscript. This ablation study section now correctly refers to Figure 7, which illustrates the role of different goal representations.
>
> 4. ### ❓W4 :The format of the pixel goal is not specified. Is it represented using special tokens or simple coordinate text?
>   - The QwenVL2.5 pre-trained model already possesses excellent visual grounding capabilities. For example, give a photo of Stephen Curry and LeBron James：
> ```text
> User: Identify basketball players and detect the key points of their heads and hands, returning them in the form of points. The primary label is the player’s name, and the secondary labels include left hand, right hand, and head.
>
> Qwen2.5-VL:  [ {"point_2d": ["394", "105"], "label": "LeBron James", "label2": "head"},     {"point_2d": ["876", "131"], "label": "Stephen Curry", "label2": "head"},     {"point_2d": ["100", "614"], "label": "LeBron James", "label2": "right hand"},     {"point_2d": ["460", "507"], "label": "LeBron James", "label2": "left hand"},     {"point_2d": ["784", "660"], "label": "Stephen Curry", "label2": "left hand"},     {"point_2d": ["945", "507"], "label": "Stephen Curry", "label2": "right hand"} ]
> ```
>   - Although its grounding performance on VLN tasks is limited, it demonstrates the ability to reason with grounded pixel locations. To fully leverage the VLM’s generalization ability, we represent pixel goals as simple coordinate text rather than using special tokens.
>
> 5. ### ❓Q3: The rationale for using a pixel goal instead of directly predicting waypoints relative to the robot.
>   - The main rationale is to minimize out-of-domain shift and catastrophic forgetting when fine-tuning the VLM on a relatively small amount of embodied data. VLMs exhibit strong abilities on 2D image data, but their understanding of real 3D space is limited.
>
>   - For example, QwenVL can accurately ground objects in an image, but it performs poorly when asked about properties such as an object’s size (in meters) or its physical coordinates from the current viewpoint, which also requires understanding the ego-view coordinate system. Directly fine-tuning System 2 to predict point goals in 3D space is not only prone to overfitting to the training scenes, but the prediction itself is extremely challenging, so the resulting performance would likely be poor.
>
>   - Notably, the community is increasingly focused on evaluating VLMs’ spatial intelligence, as exemplified by benchmarks such as VSI-Bench. Once VLMs demonstrate strong spatial reasoning, we anticipate that directly predicting real-world waypoints will become feasible.

---

> > ### Comment · Reviewer_EsMj · 2025-11-27
> > **Response to authors**
> >
> > Thanks for the rebuttal, which addresses my concerns. I will increase my rating to 6.

---

> > > ### Author Response · Authors · 2025-11-27
> > >
> > > We are pleased that our responses successfully addressed the reviewer's concerns and thank you for the valuable feedback.

---

### Official Review · Reviewer_r9jt · 2025-11-01

**Soundness:** 2
**Presentation:** 3
**Contribution:** 3
**Rating:** 4
**Confidence:** 5

**Summary:**

This paper proposes **DualVLN**, a **dual-system foundation model for vision-language navigation (VLN)**. The core insight is to decouple high-level semantic reasoning (System 2) from low-level control (System 1) in a slow–fast, hierarchical architecture.
System 2 (“Ground Slow”) is a **7 B-parameter vision-language model (VLM)** that performs mid-term waypoint planning through **pixel-goal grounding** and **self-directed view adjustment**.
System 1 (“Move Fast”) is a **multi-modal diffusion transformer policy** that conditions on both explicit pixel goals and latent features from System 2, enabling high-frequency (30 Hz) trajectory generation and dynamic obstacle avoidance.

The paper introduces a new **Social-VLN benchmark** to test navigation under dynamic, human-like obstacles, and evaluates DualVLN extensively across **VLN-CE**, **VLN-PE**, and real-world robotic platforms (wheeled, quadruped, humanoid).
DualVLN achieves new state-of-the-art performance on VLN-CE and VLN-PE and demonstrates real-time adaptability and strong cross-embodiment generalization.

**Strengths:**

**1. Clear and Novel Architectural Concept – Dual-System Slow/Fast Design:**
Decoupling a slow, deliberative VLM planner from a fast, diffusion-based controller is an elegant and well-motivated solution to the latency and fragmentation issues of end-to-end VLN models.
The “Ground Slow, Move Fast” principle is conceptually intuitive and technically grounded, providing a bridge between symbolic reasoning and reactive control.

**2. Solid Methodological Design and Implementation:**
The formulation of **pixel-goal grounding** and **latent-goal extraction via learnable queries** is novel.
System 1’s **flow-matching diffusion objective** (Eqns 1–3) and **multi-modal conditioning pipeline** (Q-Former + ViT + DiT) are clearly described and reproducible.
The asynchronous inference pipeline (2 Hz for System 2, 30 Hz for System 1) demonstrates engineering maturity and real-time feasibility.

**3. Comprehensive Empirical Evaluation:**
DualVLN surpasses all prior methods—particularly **StreamVLN** and **NaVILA**—on both **R2R-CE** and **RxR-CE** unseen splits (SR ↑ 64.3%, SPL ↑ 58.5%), and maintains superiority under physical dynamics (VLN-PE).
Zero-shot real-world tests across three robot platforms validate strong sim-to-real transfer.

**4. Introduction of Social-VLN Benchmark:**
Extending VLN to dynamic, human-populated settings (with humanoid agents) is a meaningful contribution. The proposed **Human Collision Rate (HCR)** metric provides a new perspective on social safety and trajectory recovery.

**5. Careful Ablations and Analysis:**
Ablation studies (Figure 6, Table 4) effectively dissect the roles of pixel goals and latent goals, showing tangible drops when each is removed.
The **data-scaling analysis** for System 1 provides insights into data efficiency, demonstrating near-saturation at ≈ 10% of training data.

**6. Strong Real-World Validation:**
Results on TurtleBot 4, Unitree Go2, and G1 confirm robust transfer and low navigation error, with detailed qualitative analyses (Figure 5).
Few VLN papers provide this level of real-robot evaluation.

**7. Writing and Clarity:**
The paper is clearly organized, with high-quality figures (Figures 1–2, 5–7) that make the dual-system workflow and results intuitive.

**Weaknesses:**

**1. Conceptual Novelty Is Weak and Superficial.**
The “dual-system” framing (slow = planner, fast = controller) has been explored extensively in prior works on hierarchical RL, slow–fast control, and modular VLA models (e.g., Helix 2025, Hi-Robot 2025, RoboPoint 2025). The present paper mostly *repackages* a conventional high-level planner + low-level policy pipeline with new terminology (“Ground Slow, Move Fast”) rather than introducing new algorithmic insights. The design feels more like narrative framing than a fundamental contribution.

**2. Limited Theoretical Insight:**
While the framework presents an appealing “fast–slow” dual-system design, the paper offers limited theoretical grounding for how latent-goal conditioning from the VLM stabilizes or aligns representations within the DiT-based controller. The motivation for using latent queries as a bridge between perception and action remains heuristic rather than principled. A more formal analysis, e.g., how latent conditioning improves stability, credit assignment, or planning coherence, would considerably strengthen the work’s theoretical foundation.

Furthermore, the central problem this paper addresses appears to be the offline–online training gap in modular VLN systems, similar to how ETPNav bridges DuET, ScaleVLN bridges HAMT, GridMM & BEVBert, etc., bridge their methods into the continuous environment through waypoint models and online finetuning. It is unsurprising that decoupling the LLM (planner) and controller (executor), then finetuning the latter for error correction, leads to improved downstream performance. However, this design also highlights a deeper limitation: the “fast” and “slow” systems do not jointly plan toward a shared latent goal. Instead, the fast system functions primarily as an error-corrective compensator for the slow system rather than as a hierarchically consistent planner. This undermines the claimed cognitive analogy to hierarchical reasoning and weakens the novelty of the dual-system framing.

**3. Missing Discussion of the System Mismatch.**
System 2 predicts **explicit pixel goals**, while System 1 consumes **latent goals**. The paper lacks explanation:

* Why the latent goal is preferable to directly using the pixel goal;
* How consistent or accurate the pixel-goal grounding is;
* Why System 1 actually benefits from the latent embedding.

Although L428-L431 try to explain this, it is far from convincing. To me, the results in Figure 6 and Table 4 show that it is necessary to compensate for the offline training gap for a large system during online inference. The diffusion policy could compensate for such a discrepancy by converting the action space into a continuous trajectory and correcting suboptimal pixel goals (by using a latent goal instead). However, these hypotheses are lacking evidence in the paper; no error analysis for why LLM + iPlanner/ DP are inferior, and what error system 2 corrects.


**4. Clarity of Data Generation and Training Pipeline:**
Details of **System 2 training data**, the projection from 3D to 2D pixel goals, and the **view-adjustment supervision** are somewhat compressed.
More transparency (e.g., dataset size, sample generation rate) would aid reproducibility.

**5. Lack of Real Justification for the “Fast” System.**
The argument that System 1 must operate at 30 Hz is not substantiated with latency or performance ablation. The experiments show improved results but do not prove that *asynchrony* itself yields the gains. No evidence is given that the fast controller corrects or compensates for sub-optimal pixel goals.

**6. Incremental Empirical Improvement.**
Although numbers on VLN-CE/VLN-PE slightly surpass baselines, the margin is modest and could easily stem from larger model capacity (7 B QwenVL backbone) or training data rather than the proposed dual-system design.

**7. Limited Analysis and Interpretability.**
There is no diagnostic evaluation of the intermediate goals, no visualization of latent features, and no discussion of why the dual representation improves reasoning. The Social-VLN benchmark, while new, does not isolate the contribution of the architecture. It simply provides another testbed.


**8. Potential Compute and Accessibility Concerns:**
Although System 1 is lightweight, System 2 uses a 7 B VLM (QwenVL-2.5), which may limit accessibility.
A discussion on inference cost, scalability to smaller models, or open-source readiness would be beneficial.

**Questions:**

1. Why is a **latent goal** required at all if explicit pixel goals already encode spatial intent?
2. How does System 1 benefit from the VLM’s latent state? Can you show feature alignment or ablation?
4. What empirical evidence supports the need for a **“fast” 30 Hz controller** rather than standard synchronous inference?
5. Is there any analysis showing that the **asynchronous design** contributes more than simply using a smaller local policy?
6. **System 2 → System 1 Coupling:** How are temporal inconsistencies handled when latent goals become outdated during fast inference? Is there a re-synchronization or interpolation strategy beyond using the last frame?
7. **Self-Directed View Adjustment:** How is the view-adjustment policy trained—supervised with labeled actions or via reinforcement from coverage gains?
8. **Benchmark Generalization:** Can Social-VLN agents interact with multiple moving agents simultaneously, or only single humanoid trajectories?
9. **Failure Modes:** What are common failure cases in dynamic scenes (e.g., oscillation, stuck behaviors)?
10. **Scalability:** Could System 1 be extended to 3D point-cloud inputs or depth channels without retraining System 2?
11. **Resource Footprint:** What is the average inference latency per navigation cycle (end-to-end) and GPU memory footprint during deployment?

---

> ### Author Response · Authors · 2025-11-21
> **Response to Reviewer r9jt (1/5)**
>
> 1. ### ❓W1: Conceptual Novelty Is Weak and Superficial.
> - ### 🔍 Clarification about the Difference between Dual-system (slow-fast) VLA and Modular VLA.
>   - Dual-system VLA was first proposed in robotic manipulation, as seen in works such as LCB[1], DP-VLA[2] and Helix. In this paradigm, the slow system's role is to encode the instruction and video context into an informative latent goal representation using a Large VLM. **This latent feature is trained end-to-end with the fast system, and therefore does not correspond to any explicit symbolic subgoal.**
>   - In contrast, modular VLA approaches rely on a VLM solely to parse the open-ended instruction into **explicit symbolic goals** (e.g., Hello Robot predicting language subgoals such as “pick up/place the object”, RoboPoint predicting ultimate affordance points), which are then executed by a separate policy model such as Pi0. These symbolic decompositions introduce **additional hand-designed interfaces** and therefore tend to be **more brittle**, as their performance heavily depends on the **correctness and completeness of the intermediate symbolic parsing**.
>   - Empirically, dual-system VLA consistently outperforms modular VLA, as the latent goal —learned jointly from language, history video, and current observation—captures richer and more nuanced contextual information than designed symbolic goals.
>
>    [1] Shentu, Yide, et al. "From llms to actions: Latent codes as bridges in hierarchical robot control."
>              2024 IEEE/RSJ International Conference on Intelligent Robots and Systems (IROS). IEEE, 2024.
>
>    [2] ByungOk Han, Jaehong Kim, and Jinhyeok Jang. A dual process vla: Efficient robotic manipulation
>              leveraging vlm. In Conference on Robot Learning (CoRL), 2024.
>
> - ### 🔍 Regarding the concern that “dual-system” is not a novel framing:
>   - We do not claim novelty in the dual-system framing itself. Our motivation is that existing end-to-end VLA models (7B) which map vision–language inputs directly to short-horizon actions have clear weaknesses in real robots, including unnatural motion, high latency, and poor generalization—precisely the issues a dual-system approach can mitigate.
>   - Moreover, prior dual-system VLA manipulation methods usually train both systems end-to-end with latent, non-interpretable intermediate goals. In such setups, System 2 (the VLM) is updated through gradients from System 1's  low-level action space, often **causing catastrophic forgetting and weakening generalization of VLM.**
>   - In contrast, (1) We introduce an explicit pixel-grounded intermediate goal that **narrows the finetuning gap**. (2) The progressive two-stage training of DualVLN ensures both generalized pixel goal grounding and robust trajectory prediction with parameter-efficient finetuning of powerful latent representation.

---

> ### Author Response · Authors · 2025-11-21
> **Response to Reviewer r9jt (2/5)**
>
> 2. ### ❓W2: Limited Theoretical Insight
> - ### 🔍 The motivation for using latent queries as a bridge between perception and action.
>   - Current state-of-the-art VLA models—whether end-to-end (e.g., Pi0) or dual-system (e.g., Helix)—share the same core principle: leverage large VLMs to encode visual context and language instructions into latent representations that guide the action model. The strength of a VLM lies in its pretrained multi-modal representations, not merely in its text outputs acting as a high-level planner.
>   - Our DualVLN pipeline also follows this principle:
>      ```text
>     instruction + history video + current RGB
>                     ↓
>                    VLM
>                     ↓
>     implicit multi-modal goal representation
>             ↙                 ↘
>        llm head            System 1
>              ↓                  ↓
>       2D pixel goal    world-coordinate trajectory
>      ```
>     DualVLN adopts a progressive training paradigm, by first finetuning the VLM to ground pixel goals and then performing end-to-end joint training to predict the trajectory.
>
>     The latent query serves as a parameter-efficient adaptation module—similar in spirit to LoRA—that finetunes VLM latent representations without distorting the VLM’s embedding space.
>
>     Importantly, this does not change the overall principle where the VLM encodes instruction, history, and current observations into informative goal representations that System 1 converts into executable actions.
> - ### 🔍 Regarding the concern that "It is unsurprising that decoupling the LLM (planner) and controller (executor), then finetuning the latter for error correction, leads to improved downstream performance. The fast system functions primarily as an error-corrective compensator for the slow system rather than as a hierarchically consistent planner. "
>   - The two targets describe the navigation goal from different dimensions:
>     - Pixel goals only specify an endpoint on 2D image-plane that discard depth, scale, and obstacle-avoidance information.
>     - Trajectories define a full obstacle-aware navigable path in the robot’s world-coordinate frame
>   - Their learning difficulty is progressive. DualVLN also adopts a progressive training paradigm, **which naturally produces a hierarchically consistent planner**. Stage 1 builds a strong representation foundation with abstract 2D supervision; Stage 2 adapts it to the challenging obstacle-aware trajectory prediction via parameter-efficient finetuning. A shallow analogy is the common practice in vision: first pretrain a vision backbone on image classification and then finetune it for bounding box regression in object detection.
>   - The fast system is not performing error correction for the slow system, but based on it, generates obstacle-aware, world-coordinate trajectories.
> - ### 🔍 The “fast” and “slow” systems do not jointly plan toward a shared latent goal.
>   - We want to clarify that the fast and slow systems **jointly plan toward a shared latent goal**.
>   - **In the updated paper version L511-528**, we added an ablation analysis about the consistency between Pixel Goal and Trajectory. To verify that trajectory prediction are strongly guided by the pixel goal, we analyze their consistency by projecting the predicted trajectory points onto the image plane. Using 1000 random samples from DualVLN models with different success rates on the VLN-CE benchmark, we compute two metrics: the pixel distance between the projected trajectory and the pixel goal, and their average angular deviation. **As shown in Figure 11**,  most points are concentrated in the lower-left region of the plot, indicating that the trajectories are oriented toward the pixel goal and reach areas near the pixel goal.

---

> ### Author Response · Authors · 2025-11-21
> **Response to Reviewer r9jt (3/5)**
>
> 3. ### ❓W3:Missing Discussion of the System Mismatch.
> - ### 🔍 Regarding the concern that Why the latent goal is preferable to directly using the pixel goal? Why System 1 actually benefits from the latent embedding?
>   - Explicit pixel goals on images are intuitive and easy for humans to interpret, we can navigate to the exact point without additional information. However, this does not hold for training models. Existing local planners—whether map-based methods or end-to-end PointNav policies like iPlanner or NavDP—depend heavily on depth and localization (rgb is often discarded), and their performance drops sharply when either is inaccurate.
>   - Therefore, pixel-goal grounding serves only as an auxiliary task to help the VLM develop human-like understanding of the navigation task. We still need to further end-to-end finetune the VLM representation to produce obstacle-aware trajectories and also to deal with real-time observation for asynchronous trajectory updating ability.
>   - To validate this, we modify our pipeline to:
>     ```text
>     instruction + history + current RGB           current RGB + pixel goal heatmap
>                    ↓                                            ↓
>                   VLM                                       System 1
>                    ↓                                            ↓
>               pixel goal                              world-coordinate trajectory
>     ```
>   - We add an extra pixel goal encoder (ViT) to process the RGB image plus a heatmap of the pixel goal. As expected, performance deteriorated substantially, confirming that purely visual pixel-goal navigation is challenging without a strong multi-modal context encoder.
>     | | SR (%) | SPL (%) | OS (%) | NE (m) |
>     | :--- | :--- | :--- | :--- | :--- |
>     | System2+pixel goal nav policy | 56.4 | 48.9 | 66.4 | 4.88 |
>     | DualVLN | **64.3** | **58.5** | **70.7** | **4.05** |
> |  |
> - ### 🔍 How consistent or accurate the pixel-goal grounding is.
>   - Pixel-goal grounding is not supervised with a strictly defined or uniquely correct target. There is a range of pixel locations that would all represent a “correct” pixel goal for a timestep. Therefore we just use a heuristic rule: the farthest visible trajectory point on current image.
>   - Navigation is an inherently sequential decision process, shaped by the interplay of view-adjustment actions, pixel-goal grounding, and trajectory prediction. The agent’s viewpoint changes continuously, prediction errors accumulate, and visibility varies over time. Consequently, there is no well-defined “ground-truth pixel goal” for every timestep during evaluation.
>   - Importantly, we already demonstrate that the predicted trajectory is strongly correlated with the pixel-goal grounding output, meaning that the quality of pixel-goal grounding is directly reflected in navigation metrics.
> - ### 🔍 No error analysis for why LLM + iPlanner/ DP are inferior, and what error system 1 corrects.
>   - **In the updated paper version L483-498**, we explained why LLM + iPlanner/ DP are inferior, and what error system 1 corrects:  System 1 exhibits strong vision-based obstacle-avoidance behavior in both our simulation and real-world experimental analysis. This makes it robust to slight pixel-goal regression errors that still indicates the correct direction but may place the goal near or on an obstacle, rather than actively correcting them. Crucially, this robustness does not extend to large or semantically incorrect pixel goals. **In Figure 8**,  we provide some qualitative analysis to illustrate this behavior.
>   - In contrast, point-goal navigation directly projects the pixel goal into a world-coordinate point, making it highly sensitive to even small visual prediction errors. Minor pixel deviations can result in large real-world displacements, so localization-dependent point-goal policies (e.g., iPlanner, NavDP) are less robust than System 1 and  perform worse than DualVLN.

---

> ### Author Response · Authors · 2025-11-21
> **Response to Reviewer r9jt (4/5)**
>
> 5. ### ❓W5&Q3&Q4&Q5: Lack of Real Justification for the “Fast” System.
> - ### 🔍 System 1 must operate at 30 Hz is not substantiated with latency or performance ablation. The experiments show improved results but do not prove that asynchrony itself yields the gains. What empirical evidence supports the need for a “fast” 30 Hz controller rather than standard synchronous inference?
>   - We expect System 1 to operate at high frequency (30 Hz) to generate real-time actions in dynamic scenes, rather than solely to improve navigation metrics.
>   - For example, in a crowded hallway with unpredictable moving pedestrians, System 1 must perceive the changing environment in real time and adjust velocity and heading to avoid collisions; even small delays could lead to bumps or abrupt stops.
>   - If System 1 (40M) were synchronous with System 2 (7B), each trajectory update would require a much slower pixel goal update from the 7B VLM.  In fact, in static scenes (VLN-CE), asynchronous and synchronous inference perform similarly as shown below:
>     | | SR (%) | SPL (%) | OS (%) | NE (m) |
>     | :--- | :--- | :--- | :--- | :--- |
>     | DualVLN-sync | 64.1 | 58.0 | 70.7 | 4.11 |
>     | DualVLN-async | **64.3** | **58.5** | **70.7** | **4.05** |
> |  |
> - ### 🔍 Why asynchronous design contributes more than simply using a smaller local policy?
>   -  Asynchronous inference is not specific to DualVLN; it only requires that the goal update frequency be lower than the trajectory update frequency. The usage of other local policies such as iPlanner and NavDP in Table 4 is also asynchronous.
>   - However, existing end-to-end VLA models with 7B VLM backbones adopt a single-system design, where planning and control share one forward pass, making asynchronous inference infeasible. As discussed, DualVLN benefits from both end-to-end training and dual-system architecture, achieving greater flexibility and robustness without relying on depth, odometry, or explicit pixel goals.
> - ### 🔍 How are temporal inconsistencies handled when latent goals become outdated during fast inference?
>   - System 1 receives the latent goal and the paired observation at time `t`, together with the real-time observation at time `t + k`. It then predicts a trajectory from the current state (`t + k`) toward the previously generated goal.
>   - During training, the model is exposed to extensive asynchronous tuples of this form—where goals and observations are intentionally offset in time. So both the latent query and System 1 learn to be robust to such temporal mismatches.
> 6. ### ❓W6: Incremental Empirical Improvement
> - ### 🔍 The performance margin is modest and could easily stem from larger model capacity (7B QwenVL backbone) or training data rather than the proposed dual-system design.
>   - All our baseline VLA models also use 7B VLM backbones: NaVid (Zhang et al., 2024), MapNav (Zhang et al., 2025b), NaVILA (Cheng et al., 2025), UniNaVid (Zhang et al., 2025a), and StreamVLN (Wei et al., 2025). DualVLN achieves substantial improvements over SOTA model StreamVLN using the same training data recipe, with a 7.4% absolute gain in Success Rate (SR) and a 6.6% absolute gain in Success weighted by Path Length (SPL).
> 7. ### ❓W7: Limited Analysis and Interpretability.
> - ### 🔍 There is no diagnostic evaluation of the intermediate goals, no visualization of latent features, and no discussion of why the dual representation improves reasoning.
>   - We believe we have addressed this concern for now.
> - ### 🔍 The Social-VLN benchmark, while new, does not isolate the contribution of the architecture. It simply provides another testbed.
>   - The Social-VLN benchmark highlights the specific advantages of our asynchronous dual-system design. In dynamic scenarios with moving pedestrians, the fast–slow hierarchy allows System 1 to react continuously to local changes while System 2 provides high-level guidance.

---

> ### Author Response · Authors · 2025-11-21
> **Response to Reviewer r9jt (5/5)**
>
> 8. ### ❓W8&Q10: Potential Compute and Accessibility Concerns.
> - ### 🔍 System 2 uses a 7B VLM (QwenVL-2.5), which may limit accessibility. A discussion on inference cost, scalability to smaller models, or open-source readiness would be beneficial.
>   - We have already reported inference costs in Section 5.2 (Real-World Experimental Setup). The 7B VLM can be replaced with models of any scale; larger models generally yield better performance, but we used 7B to ensure fair comparison with baselines.
>   - Most current VLA models, such as NaVid, UniNaVid, NaVILA, and StreamVLN, are open-source. We also plan to follow this practice and release our models on Hugging Face soon.
>
> 9. ### ❓Q7: Can Social-VLN agents interact with multiple moving agents simultaneously, or only single humanoid trajectories?
> - Yes, please refer to **Line 229-244 Figure 3** which shows typical robot-humanoid interactions like Simultaneous Multi-Human Encounter.
> 10. ### ❓Q8: What are common failure cases in dynamic scenes?
> - Generally, failures tend to concentrate in the following situations:
>   - Unexpected Close-Proximity Encounters: The robot often fails to react promptly when a human
> emerges suddenly from an occluded area at close range. A typical example is an unexpected en-
> counter at a corner or hallway intersection, where limited perceptual readiness leads to collision.
>   - Crowded Multi-Pedestrian Scenarios: In dense environments where multiple humans appear simul-
> taneously, the robot’s obstacle avoidance capability is overwhelmed. The complexity of predicting
> collective motion and planning a socially compliant path frequently leads to navigation failure.
>   - Constrained Navigation in Narrow Spaces: In confined areas such as doorways or tight corridors,
> the robot’s ability to proactively adjust its trajectory is significantly restricted. This spatial limitation
> considerably increases the risk of collision even with a single human.
> 11. ### ❓Q9: Could System 1 be extended to 3D point-cloud inputs or depth channels without retraining System 2?
>   - Thanks to the latent-query design for System 2, which uses parameter-efficient tuning, System 1 can be enhanced independently without retraining System 2 while still leveraging and adapting to its powerful goal representations. This allows flexibility to incorporate additional inputs, such as 3D point clouds or depth channels, to further improve the local planner’s obstacle avoidance and goal-reaching capabilities.

---

### Official Review · Reviewer_8eJ9 · 2025-11-02

**Soundness:** 4
**Presentation:** 2
**Contribution:** 4
**Rating:** 8
**Confidence:** 4

**Summary:**

This article proposes a dual-system framework called DualVLN for the vision-language navigation (VLN) task. DualVLN employs Qwen2.5-VL as System 2 to encode visual observations and language instructions, predicting pixel-level goals based on the current observation. In addition, System 2 introduces learnable latent queries to enhance its interaction with System 1.
System 1 is a diffusion-based model designed to generate short-horizon, high-frequency, and collision-free trajectories. System 2 is first trained through a pixel-goal grounding task and then fine-tuned on VLN datasets, while System 1 is trained using flow matching.
Extensive experiments are conducted on multiple benchmarks, including R2R-CE, RxR-CE, VLN-PE, and Social-VLN. DualVLN achieves state-of-the-art performance across all of these benchmarks, and real-world experiments further demonstrate its strong performance compared to existing baselines.

**Strengths:**

1.	The proposed dual-system framework is novel for the VLN task, offering a promising approach to handle high-level planning and low-level control separately while allowing joint optimization.
2.	The strong performance across multiple benchmarks demonstrates DualVLN’s remarkable capability and its potential to scale effectively with larger datasets.
3.	The dual-system design significantly enhances control frequency, which is crucial for real-world deployment.
4.	Real-world experiments conducted on wheeled, humanoid, and quadruped robots further highlight DualVLN’s strong cross-embodiment generalization ability.

**Weaknesses:**

1.	The authors should provide additional details about the ground-truth data collection process used to train the two systems.
2.	More methodological details should be included, particularly regarding the pixel-goal grounding component and the self-directed view adjustment mechanism.
3.	Ablation studies and further analyses on the self-directed view adjustment module should be provided to better understand its contribution and effectiveness.

**Questions:**

1.	How is visibility from the agent’s position measured when projecting trajectories onto 2D observations? The authors are encouraged to provide a detailed description of the procedure for generating pixel-grounding training samples.
2.	The authors should offer more explanation on the design of the self-directed view adjustment module, including how this ability is trained and its overall effectiveness.
3.	In the ablation study section, Line 425 references Table X, which does not exist in the article. In fact, Figure 6 illustrates the role of different goal representations, but it is not cited anywhere in the text. The authors are advised to clarify and revise this part for better consistency and readability.

---

> ### Author Response · Authors · 2025-11-21
> **Response to Reviewer 8eJ9**
>
> 1. ### ❓W1&W2: Additional details about the ground-truth data collection process used to train the two systems. More methodological details should be included, particularly regarding the pixel-goal grounding component and the self-directed view adjustment mechanism.
>   - Please first refer to the General Response.
>   - In addition, we have included several demo videos in the supplementary material to visually illustrate the complete process about how the self-directed view-adjustment and pixel-goal grounding mechanisms work collaboratively during a navigation process.
> 2. ### ❓W3 & Q2: Ablation studies and further analyses on the self-directed view adjustment module.
>   - While using pixel-goal grounding alone can partially guide turning actions, it only works when the future trajectory involves modest angular changes that can be projected onto the current observation. This restriction causes a large portion of both pixel goal and turning data to be excluded from training, and creates a highly imbalanced data distribution.
>   - As a result, the model tends to move only within the currently visible navigable area and fails to adjust its viewpoint effectively, especially when the instruction includes commands like "turn around". The results are shown in the following Table (the models are trained on only R2R-CE and RxR-CE data):
>     | | SR (%) | SPL (%) | OS (%) | NE (m) |
>     | :--- | :--- | :--- | :--- | :--- |
>     | Pixel-Goal Only | 20.7 | 19.6 | 22.9 | 9.01 |
>     | Pixel-Goal + Self-Directed View Adj. | **49.2** | **45.1** | **57.9** | **5.29** |
> |  |
> 3. ### ❓Q1: How is visibility from the agent’s position measured when projecting trajectories onto 2D observations?
>   - Specifically, when projecting the trajectory, we use the depth map together with the camera–point distance to identify which points fall within the visible region of the current view. Any trajectory point whose distance exceeds the corresponding depth value is treated as occluded and discarded.
> 4. ### ❓Q3: The ablation study section (Line 425) references a nonexistent Table X.
>   - Thanks for pointing this out. We have fixed the misreference in the revised manuscript. This ablation study section now correctly refers to **Figure 7 in updated version**, which illustrates the role of different goal representations.

---

### Author Response · Authors · 2025-11-21
**General Response: Details about the ground-truth data collection process and implementation for training the two systems.**

*We are committed to releasing all data, implementation code, and model checkpoints on GitHub and HuggingFace shortly.*

We have updated the following content in the **expanded appendix (after page 14)**.
##  Data Preparation and Training of System 2
We convert a discrete action sequence into three types of training data for System 2: **self-directed view adjustment**, **pixel-goal grounding** and **stop**.

### Discrete Ground Truth Action Set
- `0`: STOP
- `1`: Move forward 25 cm
- `2`: Turn left 15°
- `3`: Turn right 15°

Example action sequence of a VLN-CE trajectory data:
[3, 3, 3, 3, 3, 3, 3, 1, 1, 2, 1, 1, 3, 1, 1, 1, 0]

User prompt template:
```text
User: You are an autonomous navigation assistant. Your task is to <instruction>.
Where should you go next to stay on track? Please output the next waypoint's coordinates in the image.
Please output STOP when you have successfully completed the task.
These are your historical observations: <history>.
```

---

### 1. Self-Directed View Adjustment
- For timesteps where the future trajectory cannot be projected onto the current observation (e.g., timestep 0 with consecutive turn actions), the model predicts the sequence of upcoming turn actions instead of pixel goals.
- Example (timestep 0, next four actions are turns):

```text
<template>
User: Your current observation is <observation 0>
Assistant: → → → →  (corresponding to 3333)
```

- Maximum of 4 consecutive future turn actions are predicted.
- If the future trajectory still cannot project at later timesteps (e.g., timestep 4), the model continues predicting turn actions:
```text
<template>
User: Your current observation is <observation 4>
Assistant: → → →  (corresponding to 333)
```

---

### 2. Pixel-Goal Grounding
- For timesteps where at least one waypoint from the future trajectory can be projected onto the current observation, the data is used to train the pixel-goal grounding.
- The target pixel goal is chosen as the **pixel location of farthest successfully projected waypoint**.
- Example:

```text
<template>
User: <observation t>
Assistant: ↓  # The model first predicts ↓ which means “the next pixel goal is in view ”
User: <"look-down" observation t>  (optional: take a looking down action for better view of the goal)
Assistant: (234, 447)  # Pixel goal by projecting the last waypoint of the action chunk [1,1,2,1,1,3,1,1]
```
- In VLN-CE, the robot height is 125 cm. Adding a "look down" action improves pixel-goal grounding performance.
- In real-world experiments with a Go2 robot (60 cm), the "look down" action is less critical due to the lower camera height and is therefore omitted.

### 3. Stop
```text
<template>
User: <observation t>
Assistant: STOP   # For the last step, the model outputs STOP to indicate the task is completed.
```
### 4. Training
- Stage 1 — SFT of QwenVL (view-adjustment & pixel-goal &  stop)
  - Finetuning QwenVL to autonomously produce either turn-action sequences or coordinate text or stop depending on      the vision and language context.
  - hyperparameters:
    - Optimizer: AdamW
    - LR: 2e-5 (full finetuning)
    - Batch size: 128
    - Total steps: 14k
---

##  Data Preparation and Training of System 1
### 1. Smooth and Resample Discrete Action Waypoints
- Discrete waypoints w.r.t each projected action chunk are converted to generate smoother trajectory waypoints.
  - For example, a list of 8 dicrete action poses `[[x, y],...]` corresponding to  `[1,1,2,1,1,3,1,1]` is interpolated into 32 smooth trajectory waypoints.
### 2. Training
- Stage 2 — Prompt-Tuning QwenVL and Training System 1
  - Only pixel-goal grounding samples are forwarded to Stage 2 for trajectory supervision.
  - After Stage 1, freeze QwenVL weights. Only two modules are trained in Stage 2:
    - Latent queries (learnable embeddings inserted at [TRAJ] token positions): extract latent goal representations from the frozen QwenVL
    - DiT-based diffusion policy (System 1): generate smooth trajectory waypoints conditioned on the latent goal.
  - hyperparameters:
    - Optimizer: AdamW
    -  LR: 1e-4
    -  Batch size: 128
    -  Total steps: 15k

---

### Meta-Review · Area_Chair_g8At · 2026-01-11

**Summary:**

This paper proposes a dual-system architecture (similar to VLAs in other domains) where the high-level VLM performs reasoning and planning, and the low-level diffusion model outputs trajectories. There were a number of important concerns raised by the reviewers. Reviewer 8eJ9 mentioned lack of details for the trajectory projections and self-directed view adjustment (shared by other reviewers). Reviewer r9jt had a wide range of questions and concerns, with some important points being that the dual-architecture method is common in VLAs these days and this paper is more of a domain-specific (VLN) application of such work. Further, the reviewer noted a lack of principled discussion in terms of how/why latent-goal conditioning can improve representations within the diffusion-based model, how disentangling the two can overcome lack of joint training, and how the output/input mismatch between the two modules are overcome. The reviewer mentions lack of significant analysis with respect to this. Reviewer EsMj shared many of these previous concerns, and GZFu mentions that despite the paper mentioning "reasoning", there is no inference-time-compute style reasoning common today in LLMs and MLLMs.

**Reviewer Concerns:**

Concerns about some of the details lacking in the paper were addressed by adding more details and ablation studies (e.g. for the self-directed view adjustment module). Other analysis/clarification questions were answered as well. However, concerns about situating the work with respect to prior dual-system methods (Helix, Gr00t, etc.) is not as convincing. The authors mainly attack the existing methods' end-to-end latent training (rather than having something interpretable in-between) and claim "This end-to-end fine-tuning of the VLM can introduce catastrophic forgetting, reducing the VLM’s generalization ability and widening the sim-to-real gap. ". However, these claims are not well-substantiated and certainly not by strong data. While the authors did ablate omission of pixel-goal supervision, that does not show specifically the claimed catastrophic forgetting, reduced generalization, or increased sim-to-real gap. Overall, there should be a strong comparison to prior dual architectures or at least stronger variations on the proposed architecture (where some variations more strongly align with those prior VLAs developed for manipulation tasks) and evidence for each of these claims.

**Reviewer Scores:**

Reviewers 8eJ9 and GZFu had high scores (unlikely to further increase), and Reviewer EsMj mentioned raising their scores, and based on the nature of the questions and details in the rebuttal I don't see a reason to doubt this. Reviewer r9jt had a large number of concerns and questions, and while some of the rebuttal did address some of the requested details, some of the analysis/interpretability questions were not well-addressed and (as mentioned above) questions about situating the work with respect to existing works definitely was not strongly addressed. As a result, I don't believe this reviewer would increase (or decreased) their score.

Overall, this paper is somewhat borderline in contribution. Some of the rebuttal details and analysis do improve the paper, however, and the overall scores are positive.

---

### Decision · Program_Chairs · 2026-01-26

Accept (Poster)